# Selection, Preparation and Application of Quantum Dots in Perovskite Solar Cells

**DOI:** 10.3390/ijms23169482

**Published:** 2022-08-22

**Authors:** Yankai Zhou, Jiayan Yang, Xingrui Luo, Yingying Li, Qingqing Qiu, Tengfeng Xie

**Affiliations:** 1Engineering Research Center for Hydrogen Energy Materials and Devices, College of Rare Earths, Jiangxi University of Science and Technology, 86 Hong Qi Road, Ganzhou 341000, China; 2Faculty of Materials Metallurgy and Chemistry, Jiangxi University of Science and Technology, 86 Hong Qi Road, Ganzhou 341000, China; 3College of Chemistry, Jilin University, Changchun 130012, China

**Keywords:** quantum dots, perovskite solar cells, stability, passivation effect, energy level alignment

## Abstract

As the third generation of new thin-film solar cells, perovskite solar cells (PSCs) have attracted much attention for their excellent photovoltaic performance. Today, PSCs have reported the highest photovoltaic conversion efficiency (PCE) of 25.5%, which is an encouraging value, very close to the highest PCE of the most widely used silicon-based solar cells. However, scholars have found that PSCs have problems of being easily decomposed under ultraviolet (UV) light, poor stability, energy level mismatch and severe hysteresis, which greatly limit their industrialization. As unique materials, quantum dots (QDs) have many excellent properties and have been widely used in PSCs to address the issues mentioned above. In this article, we describe the application of various QDs as additives in different layers of PSCs, as luminescent down-shifting materials, and directly as electron transport layers (ETL), light-absorbing layers and hole transport layers (HTL). The addition of QDs optimizes the energy level arrangement within the device, expands the range of light utilization, passivates defects on the surface of the perovskite film and promotes electron and hole transport, resulting in significant improvements in both PCE and stability. We summarize in detail the role of QDs in PSCs, analyze the perspective and associated issues of QDs in PSCs, and finally offer our insights into the future direction of development.

## 1. Introduction

With the growing world population and the rapid development of industrial civilization, fossil fuels such as coal, oil and natural gas are being used in large quantities [1]. Thus, the energy crisis and environmental pollution problems are increasing, making it urgent to find new alternative sources of energy [2,3]. Solar energy is inexhaustible and is the most abundant renewable and clean energy source on the planet [2,4]. Therefore, the preparation of highly efficient photovoltaic devices is undoubtedly one of the most effective ways to solve the energy crisis.

Among the many solar cells available, perovskite solar cells (PSCs) stand out for their excellent photovoltaic properties. PSCs usually consist of the electron transport layer (ETL), the light-absorbing layer and the hole transport layer (HTL) [5,6]. The ETL is generally TiO_2_, SnO_2_, phenyl-C61-butyric acid methyl ester (PCBM). The HTL is generally 2,2′,7,7′-tetrakis-(*N*,*N*-di-4-methoxyphenylamino)-9,9′-spirobifluorene (Spiro-OMeTAD), polytriarylamine (PTAA), NiO, poly(4-butylphenyldiphenylamine),poly(3,4-ethylenedioxythiophene) polystyrene sulfonate (PEDOT:PSS). Recently, PSCs have registered a maximum efficiency of 25.5% [7], which is the highest PCE in thin-film cells.

However, compared to silicon-based solar cells, the stability of PSCs in the environment is poor [8,9]. There is also a mismatch in the energy level arrangement within the PSCs, as well as severe hysteresis effects, which are very unfavorable for charge transport and the photovoltaic properties of devices [10,11,12]. In the past decade, scholars have focused efforts on optimizing the performance of PSCs. Quantum dots (QDs), materials with excellent photovoltaic properties, not only have tunable energy bands and can optimize the energy level arrangement within PSCs, but they can also passivate defects on the surface of perovskite films and facilitate the transport of electrons and holes [13,14,15]. Therefore, QDs are considered to be promising materials for optimizing PSCs.

Although the application of QDs in PSCs has been reported, the collated data lack comprehensive classification and summary. In this article, we analyze, in detail, QDs as additives in electron transport layers (ETLs), hole transport layers (HTLs) and perovskite films, and directly as electron transport materials (ETMs), hole transport materials (HTMs), light-absorbing layers and luminescent downshifting materials. More importantly, we summarize the role of QDs in PSCs and analyze the perspectives and associated issues. Finally, future research directions are proposed for the application of QDs in PSCs.

## 2. Quantum Dot Materials

QDs are semiconductors, considered QD materials with their volumetric footprint strictly restricted at the nanoscale, and are mostly prepared by thermal injection and low-temperature solution methods. They have shown great application in solar cells [16,17], photocatalysis [18,19,20,21,22] and light-emitting diodes [23,24] in the past.

Scholars applied QDs in PSCs and found that QDs can optimize the energy level arrangement, enhancing charge transport (Figure 1). They can also increase the crystallinity of perovskite films during their crystallization, thus enhancing the quality of the film. QDs have been widely used in PSCs, where they are showing great application value.

## 3. Quantum Dots as Additives in Perovskite Solar Cells

### 3.1. Quantum Dots as Additives in Electron Transport Layers

The electron transport layer (ETL) has the function of collecting and transporting electrons to the conductive glass, while effectively blocking holes from recombination. The ideal electron transport material (ETM) should have the following properties: (1)Proper alignment of energy levels to allow for the efficient extraction of electrons.(2)High electron mobility to transfer the photogenerated electrons to the external circuits faster. However, electron transport and extraction in the ETL of PSCs are not ideal, showing stability and hysteresis issues. QDs doping into ETL presented an effective method for optimizing ETM. Table 1 lists the details of the devices [25,26,27,28,29,30,31,32,33,34,35,36,37,38,39,40,41,42,43,44].

#### 3.1.1. Carbon Quantum Dots as Additives in Electron Transport Layers

Carbon is a popular building material due to its low price, high surface area and abundant resources [45]. Doping carbon materials into PSCs can significantly improve the PCE and stability. In particular, carbon quantum dots (CQDs) and graphene quantum dots (GQDs) are widely used in PSCs due to their excellent properties. CQDs are zero-dimensional carbon-based materials, with sizes less than 10 nm [46]. CQDs consist of dispersed sphere-like carbon particles. Suitable size, low cost and good biocompatibility are extremely important for applications in the fields of biology, materials and chemistry. Due to its excellent electron transfer properties, broad light absorption, ease of surface functionalization and high electron mobility, it is mainly doped into ETL such as TiO_2_, SnO_2_ and PCBM in order to improve the performance of PSCs and increase PCE.

As a QD material, TiO_2_ has a suitable energy band structure and excellent chemical properties, generally considered the material of choice for mesoporous scaffolds and compact layers for PSCs. On the other hand, TiO_2_ has the disadvantages of low electron mobility (~10^−4^ cm^2^ V^−1^ s^−1^) and low conductivity (~1.1 × 10^−5^ S cm^−1^). Under UV light, it has strong photocatalytic effects that can easily cause the decomposition of perovskite. To solve the problem of slower charge transfer, Li et al. [26] used composites of treated CQDs with TiO_2_ as ETL for PSCs in 2017, showing that CQDs can enhance electron extraction and electron coupling between the perovskite and ETL, also achieving better energy level matching (Figure 2a) and ultimately contributing to the high performance of the device, with a PCE of 18.89% (Figure 2b). Moreover, in 2017, Jin et al. [25] introduced CQDs in the mesoporous TiO_2_ (m-TiO_2_) layer in PSCs using the modified hydrothermal method to prevent the UV-induced decomposition of perovskite films. CQDs effectively converted UV light into blue light and significantly enhanced the light stability of PSCs. In total, 70% of the initial efficiency was maintained after 12 h of full sunlight, much higher than the 20% initial efficiency of the bare device. Finally, the PCE had improved by 16.4%. In 2019, Zhu et al. [27] first used quasi-spherical CQDs as additives to PCBM for inverted PSCs (Figure 2c). The CQDs not only substantially increased the electron mobility, but also effectively prevented I^−^ ions from undergoing interfacial diffusion and inhibited the decomposition of perovskite (Figure 2d), resulting in enhanced long-term device stability and a final PCE of 18.1% for the inverted PSCs.

Furthermore, in 2020, Hui et al. [29] prepared an efficient composite ETL by doping red carbon quantum dots (RCQDs) rich in carboxylic acids and hydroxyl groups into cryogenically treated SnO_2_. The electron mobility of SnO_2_ increased significantly from 9.32 × 10^−4^ cm^2^ V^−1^ s^−1^ to 1.73 × 10^−2^ cm^2^ V^−1^ s^−1^, which set a record at that time. The end result was a remarkable increase in efficiency, with PCE from 19.15% to 22.77%. In addition, RCQDs were shown to promote the formation of highly crystalline perovskite to some degree, reducing defects at the interface between the perovskite and ETL. In the same year, Chen et al. [28] designed graphitized carbon nitride quantum dots (g-C_3_N_4_ QDs), and compounded them with SnO_2_. The g-C_3_N_4_ QDs can recast the electron density distribution of SnO_2_ crystal units, thereby eliminating the oxygen vacancy reduction trap center and facilitating electron transport. The final equipment manufactured achieved a PCE of 22.13%. The device still had an initial PCE of 90% after aging at relative humidity (RH) of 60% for 1500 h, which is undoubtedly an important step for PSCs.

By introducing CQDs into the ETL, not only can the extraction rate of electrons be increased, but also the long-term stability of PSCs can be improved. The test results showed that PSCs were still effectively improved under the harsh conditions [25,26,27,28,29].

**Figure 2 ijms-23-09482-f002:**
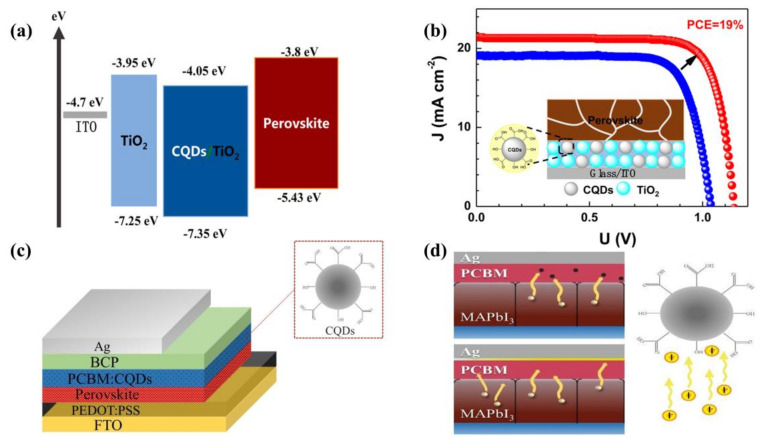
(**a**) Energy levels before and after the addition of CQDs. Reproduced from [26], with permission from the American Chemical Society, 2017. (**b**) PCE of optimal devices after CQD modifications. Reproduced from [26], with permission from the American Chemical Society, 2017. (**c**) Structure of the planar heterojunction perovskite solar cell. Reproduced from [27], with permission from the Royal Society of Chemistry, 2019. (**d**) Proposed mechanism of preventing I^−^ ions from diffusing into Ag electrode. Reproduced from [27], with permission from the Royal Society of Chemistry, 2019.

#### 3.1.2. Graphene Quantum Dots as Additives in Electron Transport Layers

GQDs are composed of single- or multi-layer graphene. Compared to graphene nanosheets with zero band gap and a low light-absorption coefficient, GQDs have some special photovoltaic properties. Examples include ultra-fast electron extraction rates (time constants of less than 15 fs), extremely long hot electron lifetimes (up to hundreds of picoseconds), broad light absorption and fluorescence properties [47].

In 2019, Ebrahimi et al. [33] synthesized GQDs by hydrothermal treatment using fenugreek leaves as a raw material. They eventually found that 2.5 vol% GQDs doped with TiO_2_-based cells had the highest PCE of 14.36%. In 2018, Shen et al. [32] synthesized GQDs using a nitric acid vapor cutting method and incorporated them onto m-TiO_2_. GQDs can effectively penetrate and cover the pores and cracks of the m-TiO_2_ film, making the modified TiO_2_ smoother. This resulted in better contact with the perovskite film, reduced the resistance inside the equipment and enhanced the charge transfer between perovskite and ETL. It was eventually found that a PCE of 20.45% could be achieved based on GQDs of size less than 5 nm. Xie et al. [30] reduced the non-radiative composite of the carrier by adding a small amount of GQDs to fill the defects present in the cryogenically treated SnO_2_ film. The photogenerated electrons in GQDs can be transferred to the conduction band of SnO_2_, enhancing the conductivity of SnO_2_, improving the electron extraction rate and reducing the complexation of the perovskite interface with ETL. The work function in light conditions dropped form 4.6 eV to 0.25 eV. The maximum steady state PCE was 20.23% and the cell had a small hysteresis.

Doping GQDs into ETL can not only boost charge transport, but also has a remarkable impact on the performance of the light-absorbing layer. In 2019, Zhou et al. [35] reported newly designed GQDs and SnO_2_ nanoparticle composites (G@SnO_2_). Compared to pure SnO_2_, G@SnO_2_ formed high-quality films and better energy level alignment on the basis of an ETL with a higher transmission rate, and then found that GQDs had the best PCE of 19.6%. In addition, the device bent at a radius of 7 nm maintained 91% of the initial PCE after 500 bending rounds (Figure 3a).

The high temperature and the long time required for the preparation of TiO_2_ limits its use for large-scale commercial production. In contrast, ETL prepared from SnO_2_ nanoparticle hydrocolloids requires a lower temperature and is more stable than TiO_2_. However, the SnO_2_ surface has defects when prepared by spin coating of the chosen SnO_2_ hydrocolloid, which can cause the decay of the perovskite absorber layer and reduce the performance of the PSCs. For this reason, Lu et al. [36] added GQDs as additives to SnO_2_ hydrocolloids in 2021. They formed compact ETL by spinning. The SEM showed that SnO_2_ with 4.5% GQDs has the flattest film morphology (Figure 3b,c), which facilitated contact with the perovskite absorber layer and reduced the internal resistance. The maximum conductivity was 5.7 × 10^5^ S/m. Thus, the PCE of the PSCs based on the optimized SnO_2_ reached 18.55%.

In the same year, Nagaraj et al. [37] used GQDs to modify SnO_2_/ZnO bilayers on ETL (Figure 3d,e). After the addition of 4% GQDs, the PCE of double ETL reached 19.81%. The open-circuit voltage (V_oc_) reached 1.17 V, and the stability greatly improved. The device retained 80% of its initial performance after 56 days in the environment with an RH of 30%~40%. This modification is simple to perform and provides a simple method for preparing PSCs close to the theoretical PCE.

GQDs can also be used to compensate for the inherent shortcomings of PCBM. PCBM has a low electrical conductivity and electron transfer rate. In particular, under continuous light, PCBM molecules undergo dimerization, a reaction that deteriorates the performance of PSCs. However, the researchers found that doping GQDs into PCBM can inhibit dimerization and improve the stability of PSCs. Yang et al. [31] obtained the best PCE by doping the PCBM films with 0.5 wt% GQDs. Compared to undoped GQDs, both short-circuit current (J_sc_) and V_oc_ were greatly improved (Figure 3f). The PCE increased from 14.68% to 17.56%, an improvement of 20%. In addition, cells with GQDs retained more than 80% of the initial PCE after more than 300 h of continuous full-spectrum sunlight exposure, with almost no degradation under air for about 30 days (Figure 3g).

Similar to the role played by CQDs, GQDs can accelerate the extraction rate of electrons and reduce the series resistance of the device, as well as effectively passivate the defects on the surface of the ETL, making the surface of the ETL smoother and more compact.

**Figure 3 ijms-23-09482-f003:**
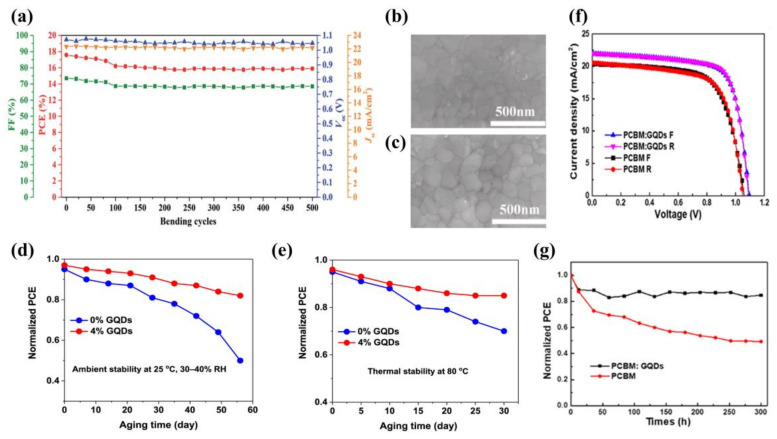
(**a**) Performance parameters as a function of bending cycles with a bending radius of 7 mm. Reproduced from [35], with permission from the Royal Society of Chemistry, 2019. (**b**,**c**) Images of 0% GQDs- and 4.5% GQDs-modified perovskite films. Reproduced from [36], with permission from Elsevier, 2021. (**d**) Stored in ambient air with RH of 30~40% and (**e**) thermal stability of PSCs kept in ambient air at 80 °C in the dark. Reproduced from [37], with permission from Elsevier, 2021. (**f**) Current–voltage (J–V) forward bias and reverse bias curves for optimal device and (**g**) device performance of doped and undoped GQDs under continuous light. Reproduced from [31], with permission from Elsevier, 2017.

#### 3.1.3. Other Quantum Dots as Additives in Electron Transport Layers

QDs as additives in ETL have positive effects on the perovskite absorber layer, in addition to optimizing electron transport. In 2017, Zeng et al. [38] optimized the PCBM with CdSe QDs in the upper part of the light-absorbing layer, which also resulted in a reduction in the roughness of the perovskite surface. The charge transport between PCBM and perovskite was also optimized, resulting in an increase in J_sc_ and FF, with a PCE of 13.73%. In 2020, Tsikritzis et al. [48] doped PC_70_BM with ultra-thin Bi_2_Te_3_ flakes. They found the optimal doping of PC_70_BM with Bi_2_Te_3_ flakes was 2% *v*/*v*, which resulted in a PCE of 18.0%. They also used Bi_2_Te_3_ flakes as the interlayer. When they formed two spin coatings of the Bi_2_Te_3_ flakes, dispersion onto the PC_70_BM led to a PCE of 18.6%. Finally, the authors combined these two engineering approaches, obtaining a PCE of 19.46%. This performance was the highest reported for inverted PSCs at that time.

ZnO is also a widely used material today, but when ZnO is used as the ETL for PSCs, it is prone to deprotonation, leading to decomposition of the light-absorbing layer. To solve this problem, in 2022, Pang et al. [42] spin-coated colloidal PbS QDs with tetrabutylammonium iodide (TBAl) on ZnO to form a ZnO/PbS-TBAl dense layer (Figure 4a). Pb^2+^ ions will synergize with the halide anion to passivate defects at the ZnO/perovskite interface. The final PCE was 20.53%, the best PCE for ZnO PSCs with MAPbI_3_ as the absorber layer (Figure 4b).

In 2021, Lv et al. [49] reported the modification of SnO_2_ using dilute CdS QDs. The crystallinity and flatness of SnO_2_ were enhanced by the modification of CdS QDs. More encouragingly, the electron mobility of SnO_2_ films doped with 1% CdS QDs is increased by an order of magnitude compared to single SnO_2_ films (from 7.34 × 10^2^ cm^2^ V^−1^ s^−1^ to 1.07 × 10^3^ cm^2^ V^−1^ s^−1^). In addition, 1% CdS QDs allowed the simultaneous optimization of the perovskite film and the SnO_2_ layer, regulating the energy level matching between the two layers and accelerating the charge transport. Not only that, but S^2−^ ions in CdS QDs will form strong coordination bonds with Pb^2+^ ions in the perovskite, establishing micro-channels that likewise speed up the charge extraction between perovskite and ETL. The final result was that the CdS (1%)-SnO_2_ based device achieved a PCE of 20.78% and the stability in air was good.

On the other hand, in 2021, Zhang et al. [40] broke with conventional thinking and introduced black phosphorus QDs (BP QDs) into both SnO_2_ and the light-absorbing layer to prepare planar-structured PSCs. The two-layer optimization strategy achieved the desired results, with passivation of SnO_2_ surface defects and an increase in conductivity in the dark case. Then they used 3-aminopropyltriethoxysilane (APTES) to surface-modify BP QDs by doping them into the light-absorbing layer. As a result of the improved grain boundaries, V_oc_ increased directly to 1.22 V. The energy band structure had also been significantly optimized (Figure 4c). The final highest PCE for PSCs optimized by two-layer BP QDs was 22.85%. In 2022, Gu et al. [41] also mixed BP QDs with SnO_2_ to prepare fluorine-doped tin oxide-coated (FTO)/SnO_2_:BP QDs/perovskite/Spiro-OMeTAD/Ag-structured PSCs. The doping of BP QDs not only had a beneficial effect on SnO_2_ by passivating its defects, but also inhibited the oxidation of the BP QDs themselves. This certainly adds to the competitiveness of BP QDs among many QDs.

SnO_2_ is generally used as the ETL for PSCs, with fewer reports on SnO_2_ QDs modifying PSCs. In 2022, Zhou et al. [43] proposed a novel idea to mix SnO_2_ QDs with m-TiO_2_ as the ETL (Figure 4d). SnO_2_ QDs not only covered the surface of m-TiO_2_, but also penetrated into its interior, effectively reducing the series resistance. The most important point was that the addition of SnO_2_ QDs inhibited charge recombination, thus minimizing the decomposition of TiO_2_ under UV light, significantly improving the stability of the device. The SnO_2_ QDs also promoted the growth of the perovskite grains to form a superior film. The PCE of the final device was 20.09%. This work provided strong support for the application of SnO_2_ QDs.

The above research provided ample evidence that QDs are effective additives to ETL to improve device stability and PCE.

**Figure 4 ijms-23-09482-f004:**
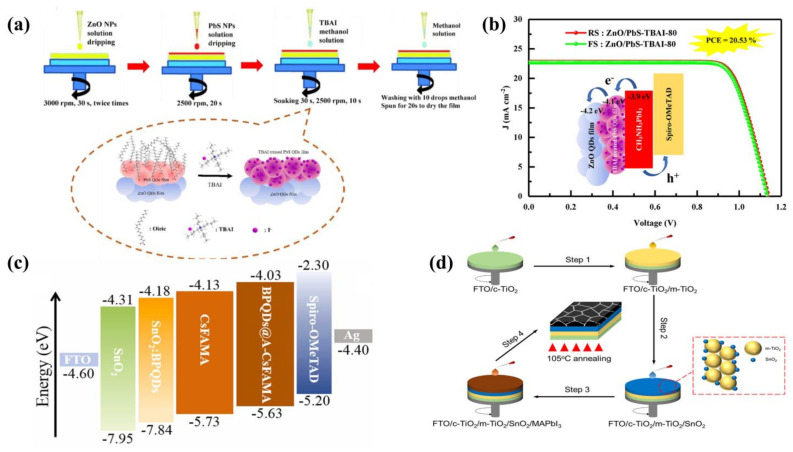
(**a**) Preparation of ZnO/PbS-TBAI ETLs and the substitution of oleic acid ligands by iodide ions. Reproduced from [42], with permission from Elsevier, 2022. (**b**) PCE of ZnO/PbS-TBAI-80-based device. Reproduced from [42], with permission from Elsevier, 2022. (**c**) Energy level arrangement of PSCs modified by BP QDs. Reproduced from [40], with permission from Elsevier, 2021. (**d**) Preparation processes of SnO_2_ QDs-modified m-TiO_2_ and MAPbI_3_ light-absorbing layer. Reproduced from [43], with permission from the American Chemical Society, 2022.

### 3.2. Quantum Dots as Additives in Perovskite Films

The light-absorbing layer is the most significant part of the PSCs and directly influences the PCE of the device. The light-absorbing layer absorbs sunlight and generates electron–hole pairs. However, a common problem with the light-absorbing layer is the inevitable occurrence of surface defects and traps during the formation of the film by perovskite crystallization. These defects not only obstruct the electron transfer of PSCs, but also allow water vapor to enter the interior of the device and decompose the perovskites, which can be fatal for PSCs. For this reason, it is necessary to introduce QDs into the perovskite absorber layer to increase the size of the perovskite grains, reduce surface defects and the non-radiative recombination of carriers, which seems to be a practical and efficient method at present. Table 2 lists the details of the devices [16,50,51,52,53,54,55,56,57,58,59,60,61,62,63,64,65,66,67,68,69,70,71,72,73,74,75].

#### 3.2.1. Carbon Quantum Dots as Additives in Perovskite Films

The merits of PSCs are largely influenced by the crystallinity and morphology of the light-absorbing layer. The doping of single-element QDs (CQDs, GQDs, Si QDs, etc.) into perovskite precursor solutions is a viable and effective strategy. In 2017, Zou et al. [50] added CQDs to a perovskite precursor solution and then prepared the light-absorbing layer using a one-step drop-coating method. They solved the problem of poor coverage of the substrate by conventional one-step solution-coating of perovskite films. The CQDs acted as heterogeneous nuclei during the perovskite crystallization process, resulting in an increase in the number of perovskite nuclei and the formation of finer grains, and better coverage of the substrate. The PCE reached 7.62%. In 2018, Wang et al. [51] synthesized nitrogen-doped carbon quantum dots (NCDs); the NCDs were rich in nitrogen and oxygen groups, which can effectively passivate the film of CH_3_NH_3_PbI_3_ and significantly reduce the recombination of carriers. In addition, the infrared spectrum showed that uncoordinated I^−^ ions on the light-absorbing layer form hydrogen bonds with nitrogen groups, and single-electron pair pyridine–nitrogen coordinated to Pb^2+^ ions, inhibiting carrier complexation, resulting in a PCE increase from 12.12% to 15.93%. Guo et al. [16] similarly reported their results on the introduction of CQDs into perovskite films. They calculated the interaction between QDs and perovskite films by density functional theory (DFT) and modeled Pb^2+^ ions with several groups, with the lowest binding energy of −2.04 eV for the Pb–carboxyl model, indicating that the QDs adsorbed on the film contribute to the stability of the system.

In another study, in 2019, Ma et al. [52] introduced CQDs into a perovskite precursor using a one-step NaOH-assisted acetone synthesis. On the one hand, they found that the carbonyl functional groups (C=O) slowed down the growth of perovskite crystals and made the grains larger. On the other hand, the carbonyl functional groups interacted with the uncoordinated Pb^2+^ ions to form the adduct CQDs-PbI_2_, which as a hetero phase nucleation site for MAPbI_3_, making the surface defects of the film passivated (Figure 5a). The final PCE reached 18.24%. This work revealed the mechanism of QDs passivation of perovskite films. In 2020, Wen et al. [53] added appropriate concentrations of CQDs to the MAI precursor solution using a two-step method. The hydroxyl and carbonyl groups on the surface of the CQDs interacted with Pb^2+^ ions, resulting in the passivation of the grain boundaries and the obtaining of large-grained perovskite film. The PCE of the final hybrid PSC was 19.17%.

Similar to the passivation effect played by QDs’ functional groups, Chen et al. [64] doped FeOOH QDs as the additive into a perovskite precursor to help grow thin films of perovskite. The iron, oxygen and hydroxyl groups strongly interacted with the uncoordinated Pb^2+^ ions and halogen ions in perovskite, validly slowing ion migration and passivating the surface defects (Figure 5b), which prevented the penetration of water and oxygen into the perovskite. Finally, the PCE based on Cs_0.05_FA_0.81_MA_0.14_PbBr_0.45_I_2.55_ was 21.0%.

In 2020, Xu et al. [54] prepared A-CQDs and CA-CQDs from two carbon sources: acetone and anhydrous citric acid. They found that perovskite films doped with A-CQDs had the largest grain size, the highest crystallinity, the lowest radiation-free composite loss and the best carrier transport performance. In addition, the hydrophobic properties of the film were also improved by the presence of alkyl groups in the A-CQDs. CA-CQDs had an inhibiting effect on the recombination of carriers in perovskite membranes, and A-CQDs had an advantageous effect on the recombination of carriers. The initial PCE of the bare device was 10.50%, while the device containing 0.05% CA-CQDs was 7.85% and the device containing 0.1% A-CQDs was 13.28%; this demonstrated that the doping of A-CQDs can improve the long-term stability of PSCs. In 2021, Li et al. [55] produced anti-solvent CQDs (ASCQDs) by the pulsed laser radiation of chlorobenzene as an additive. They adjusted the size of the ASCQDs using lasers, and ASCQDs can passivate defects on grain boundaries. The optimized luminous intensity was higher and the carrier lifetime was longer than before, demonstrating that laser-induced ASCQDs can reduce the non-radiative recombination and defect density through passivation. Finally, the PCE reached 14.95%. In addition, they also used pulsed laser radiation with ethyl acetate solvent to prepare CQDs (EACQDs) [56]. In the end, the PCE of PSC was 16.43%. Compared to conventional preparation methods, the pulsed laser radiation approach allowed the size of the QDs to be easily adjusted according to the amount of energy. It also allowed for the easy preparation of a wide range of QDs with very high purity and no ligand introduction, which set the stage for a wide application of the method in the future.

**Figure 5 ijms-23-09482-f005:**
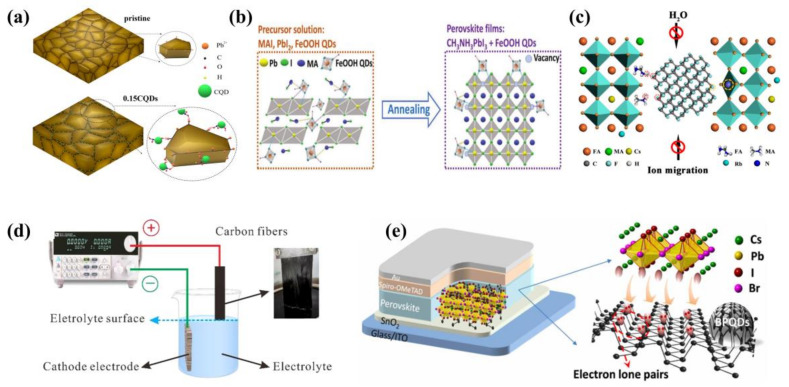
(**a**) Mechanism of CQD passivation. Reproduced from [52], with permission from the American Chemical Society, 2019. (**b**) Passivation of FeOOH QDs. Reproduced from [64], with permission from Wiley, 2019. (**c**) F-GQD-passivated perovskite film. Reproduced from [60], with permission from the American Chemical Society, 2020. (**d**) Solid-liquid interface-stripping technology for the manufacture of GQDs. Reproduced from [17], with permission from Elsevier, 2021. (**e**) Interaction between BP QDs and CsPbI_2_Br. Reproduced from [67], with permission from the American Association for the Advancement of Science, 2020.

#### 3.2.2. Graphene Quantum Dots as Additives in Perovskite Films

In 2017, Fang et al. [57] fabricated GQDs using the hydrothermal approach and introduced GQDs into the solution of perovskite precursors, and then, through one-step spin-coating, prepared perovskite films. The defects of the film were passivated by the GQDs, which reduced the series resistance and promoted charge transfer. The PCE of the device was 17.62%, an 8.2% increase in efficiency compared to the pure device. In 2018, Zhang et al. [58] added GQDs to perovskite precursors to improve the quality of the perovskite film. They obtained a highly crystalline and smooth film. The film had enhanced light absorption, which promoted charge extraction, and was increased by 11% in the PCE compared to the pure device. However, when the amount of GQDs added was increased from 0.2% to 1%, the PCE gradually decreased, which can be attributed to the inefficient separation of the photogenerated electron–hole pairs. In 2019, Subramanian et al. [76] used a one-step solvent-induced rapid deposition–crystallization technique and developed superior quality white fluorescent GQDs, which eliminated defects in perovskite film and induced effective charge transfer. The hybrid film had a two-times-higher photocurrent and a three-times-higher responsiveness than pure MAPbI_3_ film.

GQDs added through elements have been shown to result in GQDs with more active sites and tunable energy structures. In 2019, Gan et al. [59] designed graphite–nitrogen doped graphene quantum dots (GN-GQDs) as additives to perovskite film. N doping modulated the GQDs structure while maintaining good charge mobility and conductivity. In addition, the conduction band (CB) of GN-GQDs was −4.2 eV, which matched with the perovskite with CB at −3.9 eV. This can guarantee the efficient transport of electrons and holes at the grain boundary. The valence band (VB) of GN-GQDs was −6.5 eV, much deeper than the perovskite with a VB at −5.5 eV, which can efficiently reduce the recombination of electrons and holes at the grain boundary. The PCE of the PSC was 19.80%, and it can remain stable at room temperature and under an RH of 30% for about 30 days. F atoms were also doped into GQDs (F-CQDs) by Yang et al. [60] to modify GQDs (Figure 5c). The strong C–F bond made the film inert and hydrophobic, avoiding the decomposition of the perovskite film. As a result, the PCE of the flexible device doped with F-GQDs was 20.4%. The stability had also been greatly improved, maintaining 85% of the initial PCE for 600 h of continuous operation at a temperature of 85 °C filled with N_2_. In 2021, Zhou et al. [17] created an innovative solid–liquid interfacial exfoliation technique (Figure 5d), and manufactured high-quality GQDs with a carbon-brazing dimension. Further, they synthesized N-GQDs and S-GQDs using a hydrothermal method. Harmful non-radiative complexes were significantly reduced by the interaction between the GQDs and Pb^2+^ ions. For PSCs tailored to customized GQDs, the carbon electrode-based CsPbIBr_2_ PSC had a maximum PCE of 9.80%, with a great improvement in stability.

#### 3.2.3. Other Quantum Dots as Additives in Perovskite Films

Black phosphorus quantum dots (BP QDs) have also been added to perovskite films to improve the quality of films and enhance the photovoltaic performance of devices. In 2020, Gong et al. [67] reported the preparation of BP QDs by probe ultrasonic liquid stripping. The strong electronic interaction between the lone pair of electrons of BP QDs and Cs^+^ ions resulted in an improved conductivity of the device (Figure 5e), and the special core-shell structure formed between the BP QDs and CsPbI_2_Br enhanced the stability of the perovskite. In addition, they found that the adhesion energies of BP QDs to PbI_2_/CsBr and CsBr/PbI_2_ were the lowest, −1.218 eV and −2.084 eV, respectively, which were much lower than the adhesion energies attached to SnO_2_ crystals using density flooding theory DFT calculation. The final PCE of 15.47% was obtained based on a 0.7 wt% BP QD device.

In current research, metal sulfide QDs have been widely used in PSCs, particularly PbS QDs, which can absorb light in the infrared range. In 2018, Han et al. [62] creatively designed a one-step method to prepare MAPbI_3−x_ and PbS QDs hybrid precursors. The one-step strategy was simpler than the two-step strategy and retained the photovoltaic properties of the perovskite and QDs. XRD showed that the addition of too many PbS QDs can have a bad effect on perovskite crystallization. Moderate amounts of PbS QDs can promote perovskite nucleation and reduce surface defects. The band edge shifted due to the narrow band gap of the PbS QDs and the small strain at the MAPbI_3_-PbS interface, enabling the film to have an enhanced light absorption range. The PCE of the final device was 18.6%. In 2019, Ngo et al. [65] tested the performance of PSCs prepared from PbS QDs with different capping ligands including methylammonium lead iodide (MAPbI), cesium lead iodide (CsPbI) and 4-amino benzoic acid (ABA). The PbS QDs enhanced the selective orientation of the (110) crystal plane and the crystallinity of the (110) and (220) crystal plane, with the ABA ligand having the most significant effect. As for the optical properties, Figure 6a shows that the luminous intensity photoluminescence (PL) spectrum of the added PbS-CsPbI QDs was three times higher than that of the reference film and 1.5 times higher than that of the PbS-ABA and PbS-MAPbI film. Ultimately, in terms of PCE, PbS-ABA QDs based on PbS-ABA had the highest PCE, which was 18.22%. In 2020, Masi et al. [68] embedded PbS QDs into formamidinium lead iodide (FAPbI) to stabilize their black phase. The black phase was the most stable phase of the FAPbI, but it was extremely susceptible to conversion to the yellow δ phase, which was unstable during synthesis. Moreover, because PbS QDs can form high-energy strong bonds with the black phase, at a low temperature of 85 °C, a stable black phase of FAPbI can be formed within 10 min. This was the record for the lowest temperature and fastest synthetic FAPbI to date. PbS QDs maintained a low band gap of 1.5 eV after doping, avoiding the band gap blueshift phenomenon. The PCE of the final FTO/SnO_2_/FAPbI-PbS/Spiro-OMeTAD/Au device was 18.00%. The next year, Zhang et al. [73] introduced PbS QDs into Pb–Sn inorganic perovskite, which markedly suppressed the phase segregation and improved the film quality. Ultimately, the PCE of planar heterojunction PSC was 8%, and the PCE of mesoporous PSC was 6%.

In 2021, Li et al. [69] prepared WS_2_ QDs as additives to perovskite film by pulsed laser irradiation in the green anti-solvent ethyl acetate (EA); the process is shown in Figure 6b. This method is faster than conventional methods, and the result gave extremely high-purity QDs, no ligand introduction and facilitated charge transfer. Experiments showed that 0.1 mg/mL of WS_2_ QDs was the optimum concentration. When the concentration was too high, the WS_2_ QDs became the center of new trap state, increasing the trap density on the surface of the film and discouraging the charge transfer. The final maximum PCE was 16.85%.

In addition, Zhou et al. [66] first combined Si QDs and CsPbBr_3_ inverse opal (IO) in 2019 (Figure 6c). The fluorescence resonance energy transfer (FRET) between Si QDs and CsPbBr_3_ IO Si QDs emitted light that was absorbed by CsPbBr_3_, which allowed more carriers to be generated, resulting in a significant increase in solar energy utilization. The final PCE was 8.31%. This design opened up a new path for future applications of Si QDs.

S and Sn are abundant resources and environmentally friendly, so they are being exploited by researchers. Han et al. [61] embedded SnS QDs in CH_3_NH_3_PbI_3_ film using in-situ crystallization, increasing the (220) crystal plane content and crystallinity of the film. In addition, because of the low band gap of SnS QDs (1.3 eV), the hybrid film enhanced the light capture in the visible and near-infrared light range (wavelength 800–900 nm), as well as the incident photocurrent conversion efficiency (IPCE) spectra. In 2022, Ou et al. [75] added non-toxic SnS QDs that synthesized to Cs_2_AgBiBr_6_ by the non-aqueous solvothermal method. The (111) crystal plane spacing of SnS QDs was 0.284 nm, while the (400) crystal plane spacing of Cs_2_AgBiBr_6_ was 0.287 nm, which was a good match between them. The introduction of SnS QDs provided nucleation sites for the formation of film and improved the crystallinity of perovskite while reducing the density of defects. The PCE of the final non-toxic PSC obtained was 1.95%, despite the low efficiency, which was the beginning of an emerging future for non-toxic calcium titanite solar cells.

Lead-based halide PSCs are the frontrunner of photovoltaic studies, but the toxicity of Pb is a critical bottleneck for the practical use of PSCs. The potential for Pb exposure during the extensive production and operation of PSC technology requires extreme caution due to concerns to the environment and humans. So, it is necessary to eliminate or replace Pb with other non-toxic or less toxic elements that can overcome environmental issues. Among many elements, Sn-based PSCs are non-toxic and environmentally friendly, but they are criticized for their susceptibility to Sn^2+^/Sn^4+^ oxidation, making PSCs less stable and low PCE. There are many ways to inhibit the oxidation of Sn^2+^/Sn^4+^ ions, but all of them contain additives, which can introduce impurities, directly reducing the performance of PSCs. Thus, in 2021, Mahmoudi et al. [71] made the first attempt to prepare Sn-based mesoporous PSCs from a mixture of reduced graphene oxide sheets anchored with Sn quantum dots (rGO-Sn QDs) and halogenated tin perovskite in the photoactive layer. This approach effectively inhibited the oxidation of Sn^2+^/Sn^4+^ ions, and no impurities were introduced (Figure 6d). The grain size of the rGO-Sn QD-doped film was larger and more compact than the pure film, which can reduce the possibility of water vapor and oxygen entering the device. The final PCE was 7.7%. This work highlighted a new approach to overcome the limitation of environmentally friendly alternatives to Pb-based PSCs, which showed potential for the development of Pb-free PSCs.

Ti_3_C_2_T_x_ QDs are a novel material to be developed with many advantages. In 2021, Liu et al. [70] fabricated Ti_3_C_2_Cl_x_ MXenes through inorganic salt melting (Figure 6e). This method eliminated the OH end, which can react with CH_3_NH_3_^+^ ions and degraded perovskite. Then, they prepared Ti_3_C_2_Cl_x_ QDs using the hydrothermal reaction method and mechanical sonication. The Cl^−^ ions of Ti_3_C_2_Cl_x_ interacted strongly with the Pb^2+^ ions (forming the Pb–Cl bond with a binding energy of 301 kJ/mol), impeding the growth of the perovskite nuclei and increasing their crystallinity. Moreover, defects can be passivated by QDs, resulting in lower residual tensile strain in the film. In addition, compared to the conduction band (CB) of the perovskite with −4.10 eV, the CB of the modified perovskite with Ti_3_C_2_Cl_x_ QDs (−4.23 eV) was closer to the CB of the SnO_2_ (−4.31 eV), demonstrating that a better energy level alignment had been formed between the perovskite and ETL. Because of the accelerated charge transfer, the final PCE of the device was 21.31%, which was an encouraging value.

**Figure 6 ijms-23-09482-f006:**
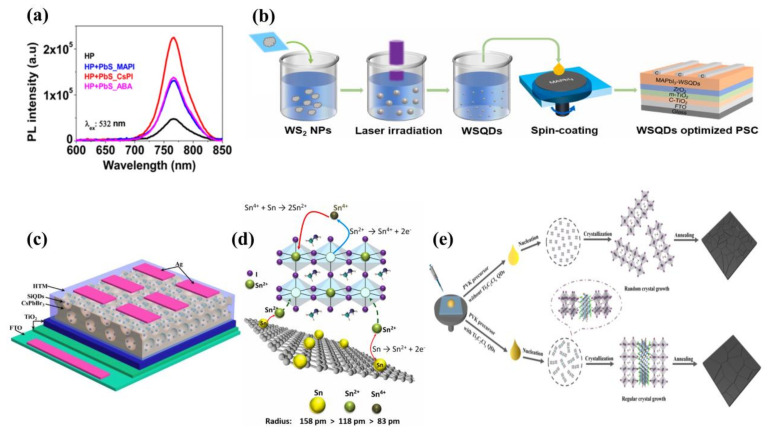
(**a**) Photoluminescence (PL). Reproduced from [65], with permission from the Royal Society of Chemistry, 2019. (**b**) WS_2_ QDs prepared by pulsed laser irradiation. Reproduced from [69], with permission from Elsevier, 2021. (**c**) Si QDs/CsPbBr_3_ IO PSC. Reproduced from [66], with permission from Elsevier, 2019. (**d**) Suppression mechanism of Sn^2+^/Sn^4+^ ion oxidation and formation of Sn^2+^ vacancies in perovskite structure. Reproduced from [71], with permission from Elsevier, 2021. (**e**) Nucleation and crystallization processes in Ti_3_C_2_Cl_x_ QDs-treated and Ti_3_C_2_Cl_x_ QDs-treated perovskite film. Reproduced from [70], with permission from Elsevier, 2021.

### 3.3. Quantum Dots as Additives in Hole Transport Layers

The hole transport layer (HTL) in PSCs serves to extract, collect and transport the holes generated by the light-absorbing layer to the metal electrode and to block the injection of electrons. It has a key influence on the photovoltaic performance and stability of PSCs. Therefore, it should have excellent cavity transport. The highest occupied molecular orbital (HOMO) energy level of the ideal HTL should be compatible with perovskite’s valence band edge energy level to minimize energy barriers. Table 3 lists the details of the devices [77,78,79,80,81,82,83,84].

#### 3.3.1. Carbon Quantum Dots as Additives in Hole Transport Layers

Currently, Spiro-OMeTAD is the most widely used HTM, but it is highly hygroscopic and tends to cause damage to perovskite films, and the energy-level mismatch with the active layer of the perovskite leads to a high loss of energy during transport [77]. It is important to improve the performance of PSCs by rationally doping Spiro-OMeTAD without affecting the stability of the device. Due to the richness of functional groups on the surface of CQDs and GQDs, they are considered to be effective additives for HTL.

In 2021, Liu et al. [77] synthesized CQDs, 4 nm in size, using the microwave strategy and applied them in Spiro-OMeTAD. The VB of Spiro-OMeTAD was −5.20 eV in the vacuum level. However, the VB of the 6% CQD-modified Spiro-OMeTAD was −5.44 eV, indicating that CQDs can adjust the band level, resulting in an increase in hole transport capacity. They can also passivate the interfacial trap state, resulting in a remarkable reduction of defects on the Spiro-OMeTAD surface. Finally, the PCE of the device was 20.41%.

After Spiro-OMeTAD was first used as HTM, NiO was also gradually widely used as the HTM for inverted PSCs due to its solution processability and high light transmission. However, there is also an energy level mismatch and low electrical conductivity of NiO. To settle these issues, in 2020, Kim et al. [79] prepared CQDs by hydrothermal reaction synthesis and added them to the NiO, reducing the current–voltage (J–V) hysteresis from 4.5% to less than 1%. As shown in Figure 7a, the PSC still maintained 70% of the initial efficiency at 192 h under atmospheric conditions. The highest PCE of this PSC was 16.91%.

#### 3.3.2. Graphene Quantum Dots as Additives in Hole Transport Layers

GQDs are also widely doped into HTL by researchers [85]. In 2017, Zhang et al. [82] used novel γ-GQDs as a surface dopant for Spiro-OMeTAD. The introduction of GQDs improved the hydrophobicity of Spiro-OMeTAD and enhanced the stability of the PSCs. The valence band energy level of the Spiro-OMeTAD doped with GQDs reduced from −5.22 eV to −5.25 eV, facilitating the collection of holes. As a result, the PCE increased from 17.17% to 19.89% compared to the same type of device without the introduction of GQDs. Zhou et al. [86] prepared the incorporation of a bilayer composed of rGO and PTAA as the HTL in planar-inverted MAPbI_3_ PSCs. As a result of the aforementioned HTL bilayer characteristics, faster hole extraction, superior quality MAPbI_3_ crystals, and better interfacial morphologies were achieved. Finally, they obtained a PCE of 17.2%. In 2020, Kim et al. [87] systematically explored rGO as the additive in perovskite and Spiro-OMeTAD, with different reduction levels (O18.42%, O9.67%, O7.14%). They found that when rGO_O9.67%_ was employed with Spiro-OMeTAD, where crystallization was effectively retarded, it can enhance the stability of the PSC. Notably, the devices with rGO in Spiro-OMeTAD demonstrated more stable performance at 85 °C, showing an ~30% PCE drop over 62 h compared to 51% for the device without rGO.

In 2020, Wang et al. [81] modified GQDs with amino groups (AGQDs) and applied them to NiO (Figure 7b). On the one hand, N atoms in the AGQDs promoted the crystallization of perovskite film through the Lewis base-acid interaction (Figure 7c). On the other hand, AGQDs optimized the energy level arrangement between NiO and perovskite, promoting the extraction of holes. The final PCE was 18.10%, which can compare to the efficiency of the world’s most advanced inverted flexible PSCs at that time.

In 2020, in order to solve the problems of carrier accumulation, low fill factor (FF) and high leakage current in HTL, Li et al. [80] prepared PEDOT:PSS/GQDs composite films using the spinning coating method and p-i-n structure. Under AM 1.5 illumination, the average PCE was 15.24% and the optimal PCE was 16.16%. The improved conductivity of the film promoted hole transport with excellent charge collection efficiency compared to the original PEDOT:PSS. Meanwhile, they detected residual PbI_2_ on PEDOT:PSS/GQDs film (Figure 7d), and the presence of PbI_2_ could effectively passivate the film and improve the quality of the perovskite film.

In addition, graphdiyne quantum dots (GD QDs) are potential materials that can be doped into HTL. Graphdiyne is a Π-conjugated material with sp^2^ and sp hybridization, and thus it has a graphene-like structure. In 2015, Xiao et al. [83] first doped GD QDs into poly-3-hexylthiophene (P3HT) film as HTL (Figure 7e). The strong Π-Π superposition between GD QDs and P3HT resulted in increased hole transfer efficiency. In addition, GD QDs exhibited unique scattering properties that significantly enhanced the long-wavelength light scattering from the HTL, resulting in increased light absorption of PSCs with a PCE of 14.58%. The stability of the device also improved. Therefore, sp^2^–sp hybrid graphdiyne has potential applications in the field of optoelectronics.

#### 3.3.3. Other Quantum Dots as Additives in Hole Transport Layers

In 2022, Zheng et al. [84] first doped oleic acid (OA)-covered PbSO_4_(PbO)_4_ QDs into Spiro-OMeTAD. Not only was its conductivity increased, but also the stability of conductivity was enhanced. Similar to the effects of other QDs, PbSO_4_(PbO)_4_ QDs enhanced the ability to extract holes from perovskite to HTL. The final PCE for the device was 22.66%. The introduction of PbSO_4_(PbO)_4_ QDs filled the blank in multi-element QDs’ doping of HTL, which is a milestone at present when only single-element QDs are generally used as additives in the HTL.

## 4. Quantum Dots as Electron Transport Materials

QDs can also be used directly as electron transport material (ETM), hole transport material (HTM) and light-absorbing layers, in addition to being used as additives and interfacial modifications. Metal oxide QDs are used by researchers as ETM for PSCs now, offering superior properties compared to current metal oxide materials. Table 4 lists the details of the devices [88,89,90,91,92,93,94,95,96,97,98].

In 2015, Ameen et al. [88] used ZnO QDs treated with an atmospheric plasma jet as ETL for flexible PSCs. They obtained a PCE of 9.73%, which although less efficient for devices, broadened the pathway for optimizing PSCs. On this basis, in 2016, Tavakoli et al. [90] prepared ZnO/rGO QDs (reduced-state graphene oxide QDs) with a core-shell structure. The special core-shell structure reduced the charge transfer resistance and improved the interfacial charge transfer efficiency. Meanwhile, rGO passivated the surface of the ZnO QDs, effectively preventing the decomposition of the perovskite film during annealing at 100 °C. The final device prepared on FTO glass substrates had a PCE of 15.2%.

In 2015, Tu et al. [89] prepared three ETMs (TiO_2_, TiCl_4_, TAA (titanium diisopropoxide bis(acetylacetonate)) by spin coating. Among them, the TiO_2_ QDs performed well, as their low series resistance and high parallel resistance resulted in faster charge transfer and a final PCE of 16.97%, much higher than the PSCs based on TiCl_4_ and TAA (Figure 8a). In 2019, Hossain et al. [99] developed a scalable low-temperature TiO_2_ ETL based on pre-synthesized crystalline nanoparticle (np-TiO_2_). Nb^5+^ doping increased the conductance through the np-TiO_2_ ETL, reduced the series resistance and improved the performance of the PSCs. The optimal PCE they obtained was 19.5%.

In 2018, Park et al. [93] prepared ligand-covered SnO_2_ QDs (Figure 8b) using the reverse micelle-water injection method. It was found that the ligands on its surface spontaneously exchanged with perovskite (Figure 8c), forming SnO_2_ QDs-MAPbI_3_ junctions, which promoted the electron transfer from the SnO_2_ QDs to the light-absorbing layer. The final PCE was 18.71%. This experiment eliminated long-chain groups from the surface of SnO_2_ QDs without the need to remove ligands, reducing the interference of impurities and carving out a new pathway for the future elimination of ligands from the surface of QDs. In 2019, Liu et al. [95] prepared excellent SnO_2_ QDs films by spin-coating ethanol precursor solutions at room temperature. This method was green, and it avoided the residual hazards of additives. Despite the average PCE being 18.80%, the hysteresis factor reached 22.3%, which was significantly higher than the comparable PSCs. For this reason, they modified the device by C_60_-sam and the hysteresis effect was significantly reduced, down to 4.7%. 

The annealing process is the most important step in the fabrication of SnO_2_ QDs and many methods have been devised by researchers to optimize the film. Examples include the strong pulsed photon annealing method [100] and the UV/ozone treatment method [101]. Although these methods have been successful in improving the quality of the film, they increase the cost of fabrication and the complexity of the process, which is not conducive to the commercialization of PSCs. To solve this problem, in 2020, Wang et al. [98] first used ethanol to vapor-treat SnO_2_ QDs during annealing, which eliminated defects on the surface of the films and led to a significant increase in the charge transport rate of the film. The final PSC showed a much higher PCE of 17.66% (Figure 8d). The method was simple, environmentally friendly and non-toxic. The properties of the films produced were comparable to those produced by other methods. In 2022, Eliwi et al. [102] designed bilayer SnO_2_ QDs ETLs. Though different tests, they found that the bilayer ETL composed of lithium-doped compact SnO_2_ (c(Li)-SnO_2_) at the bottom and potassium-capped SnO_2_ nanoparticle layers (NP-SnO_2_) at the top enhanced the charge transport properties of PSCs, and significantly reduced the degree of ion migration. The resulting optimal PCE reached 20.4%, and strongly reduced J-V hysteresis for PSCs. This research provided a good example for optimizing bilayer ETLs in the future.

PCBM is a tradition class of highly conductive material, but it is too costly to be used on a large scale. In 2017, Tan et al. [91] used pyridine-covered CdSe QDs as ETL to replace the PCBM. The thickness of the QD layer affects the charge transport. When the thickness of the CdSe QDs layer increased to 25 nm, inorganic inverted PSCs exhibited the best PCE at 14.2%. However, when the thickness was larger than 25 nm, this caused an increase in the internal series resistance, which can degrade device performance. In 2018, Fu et al. [92] used bipolar BP QDs as ETL. On the one hand, the BP QDs promoted the efficiency of electron transfer and suppressed carrier non-radiative recombination. On the other hand, the BP QDs assisted the crystallization of perovskite, generating a high-quality perovskite film, and the final PCE was 11.26%. Although currently non-metallic oxide QDs are less efficient when used as ETL, non-metallic QDs will perform even better as ETM in the future.

Compared to bulk materials, QDs have a tunable band gap and they are able to optimize energy level arrangements. Therefore, QDs have great potential for applications as ETM.

**Figure 8 ijms-23-09482-f008:**
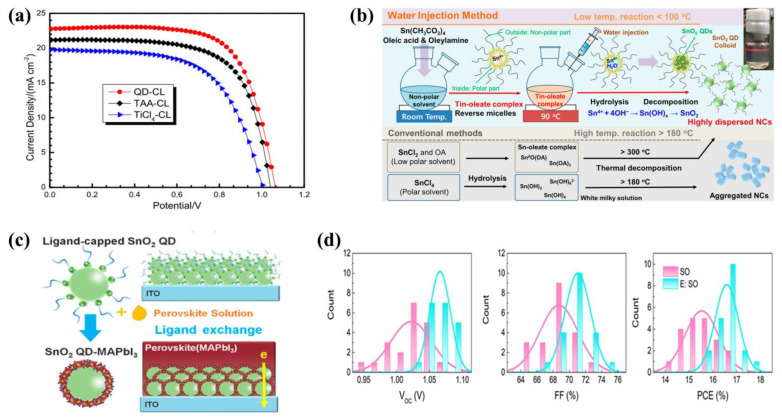
(**a**) J-V curves of the PSCs based on QDs, TAA and TiCl_4_. Reproduced from [89], with permission from the Royal Society of Chemistry, 2015. (**b**) Synthesis of SnO_2_ QDs and schematic design of colloidal solutions. Reproduced from [93], with permission from the American Chemical Society, 2018. (**c**) Ligand exchange during deposition. Reproduced from [93], with permission from the American Chemical Society, 2018. (**d**) V_oc_, FF and PCE of optimal device treated with ethanol vapor. Reproduced from [98], with permission from Elsevier, 2020.

## 5. Quantum Dots as Hole Transport Materials

Among the current hole transport materials (HTMs), Spiro-OMeTAD and PEDOT:PSS are the two most widely used. However, when Spiro-OMeTAD is used as the HTM, additives have to be added. The use of additives increases the manufacturing costs and makes Spiro-OMeTAD hydrophilic. PEDOT:PSS is inherently hydrophilic and is highly susceptible to absorb atmospheric moisture to reduce the stability of PSCs, in addition to its high acidity, which tends to corrode ITO. Therefore, the search for new HTMs is urgent. QDs have been adopted as an HTM in PSCs because of their high hole transport rate and their ability to optimize energy level alignment. Table 5 lists the details of the devices [103,104,105,106,107,108,109,110,111,112,113,114,115,116,117,118,119,120,121,122,123,124,125,126].

In 2015, Hu et al. [105] used PbS QDs as an HTM and controlled the size of the PbS QDs to obtain different PCE. The device with an energy band of 1.4 eV and an absorption peak of 890 nm had the best PCE of 7.5%. Li et al. [106] synthesized OA-covered PbS QDs by thermal injection, which retarded the charge recombination and ultimately obtained a PCE of 7.88%. However, there was a large gap between the PCE of 11.11% when Spiro-OMeTAD was used as an HTM.

In 2016, Paulo et al. [110] used a polymerization–carbonization hydrothermal method to prepare CQDs (Figure 9a) and adopted it as an HTM. Because of the poor coverage of perovskite on the m-TiO_2_ surface, this resulted in a device with only 3% PCE at that time. In another study, the HTM with high cost and a complex structure was substituted with CQDs by Kasi Matta et al. [127]. Good hole transfer efficiency was found between the light-absorbing layer and the CQDs. CQDs have been widely used in ETL and perovskite films, but relatively little has been reported in terms of use as an HTM, and they are necessary to enhance future research about the application of CQDs as HTMs.

Ternary QDs have also now been attempted by researchers for applications in PSCs. In 2019, Zhang et al. [118] found that PSCs based on CuInSe_2_ QDs had an improved hole extraction capacity, with a final PCE of 12.80%. When they used Spiro–OMeTAD as the HTM, the device had the highest V_oc_ and J_sc_ and a PCE of 15.6%. There is still a gap between CuInSe_2_ QDs and Spiro-OMeTAD, but CuInSe_2_ QDs as an HTM can improve the stability of the PSC (Figure 9b). After this, Kim et al. [113] used hydrophobic oleylamine (OAm) ligand-covered CuIn_1.5_Se_3_ QDs as an HTM, and the hydrophobic OAm on the surface of the QDs effectively prevented water vapor from penetrating into the perovskite and inhibited its decomposition (Figure 9c), resulting in a PCE of 13.72% for an area of 0.12 cm^2^. The stability of the device was improved, showing an excellent PCE retention of 89.2% after 30 days relative to its initial value at RH of 25%.

Cu_12_Sb_4_S_13_ QDs have been extensively studied for their suitable valence band positions. In 2018, Tamilselvan et al. [114] prepared oleic acid-covered Cu_12_Sb_4_S_13_ QDs with a resistivity of only 0.043 Ω cm, much lower than that of conventional HTMs, with the resistivity of 4000 Ω cm for the most commonly used Spiro-OMeTAD. The low resistivity gave the device a high electrical conductivity with a final PCE of 6.5%. In 2019, Liu et al. [122] prepared Cu_12_Sb_4_S_13_ QDs with different sizes and found experimentally that the QDs of size 5.7 nm performed best, obtaining a PCE of 14.13%. Due to having the most suitable energy band alignment, compared to the Spiro-OMeTAD based device, J_sc_ increased from 20.73 mA cm^−2^ to 21.85 mA cm^−2^. In 2020, Liu et al. [124] continued their research on Cu_12_Sb_4_S_13_ QDs to optimize the performance of all-inorganic PSCs by changing the surface properties of the QDs through ligands exchange (Figure 9d). The replacement of surface OAm by 3-mercaptopropionic acid (MPA) reduced the band gap and increased the hole migration rate. The amount of MPA affected the performance of the device—the highest PCE, 10.02%, was obtained when the MPA:Cu_12_Sb_4_S_13_ QDs was 3:5. The inorganic PSCs maintained 94% of the initial PCE after 360 h in air, whereas the Spiro-OMeTAD-based cell was unable to maintain its stability in air and its performance decreased rapidly, with only 42% of its initial PCE after 360 h.

The quaternary semiconductor Cu_2_ZnSnS_4_ QDs were first applied to PSCs in 2015 as a material with a high optical absorption coefficient and a suitable band gap (1.5 eV). Wu et al. [108] used Cu_2_ZnSnS_4_ QDs as an HTM, and the device showed significantly optimized light absorption in the visible region from 400 to 800 nm, ultimately resulting in a PCE of 12.75%. In 2016, Yuan et al. [111] introduced Cu_2_ZnSnS_4_ QDs into PSCs, and then replaced the S atoms with Se atoms to obtain a better energy band alignment. However, due to the relatively shallow valence band of Cu_2_ZnSnS_4_ QD devices, there was a decrease in PCE from 10.72% to 9.72%. In 2019, Zhou et al. [123] obtained 4.84% PCE for all-inorganic CsPbBr_3_ PSCs after using Cu_2_ZnSnS_4_ QDs as the HTM, with significant higher extraction and hole transport rates.

**Figure 9 ijms-23-09482-f009:**
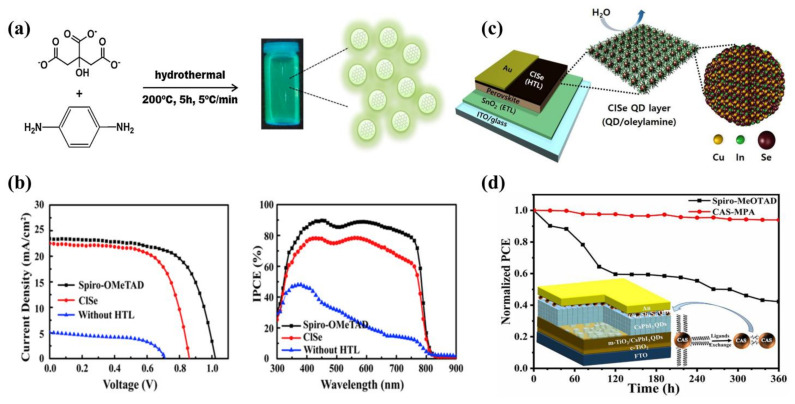
(**a**) Hydrothermal synthesis of CQDs. Reproduced from [110], with permission from Elsevier, 2016. (**b**) J-V curves during champions with different HTM and IPCE spectra. Reproduced from [118], with permission from Elsevier, 2019. (**c**) PSC based on CuIn_1.5_Se_3_ QDs and hydrophobic oleylamine, preventing water from entering perovskite. Reproduced from [113], with permission from Elsevier, 2019. (**d**) PCE of all-inorganic PSCs after ligand exchange. Reproduced from [124], with permission from the American Chemical Society, 2020.

Cu_2_O-based PSCs are generally inverted structures, while few PSCs have been reported for n-i-p-type structures. In 2019, Liu et al. [117] used Cu_2_O QDs as HTM for n-i-p-type PSCs. They modified the surface of Cu_2_O QDs with ethenyltriethyloxysilane, introduced hydrophobic groups to improve the hydrophobicity of PSCs, and eventually obtained a PCE of 18.90%, which was the highest efficiency for Cu_2_O-based PSCs at that time.

Compared to conventional HTMs, the PCE of PSC-based QDs as HTMs is significantly lower, which is the shortcoming of QDs as an HTM. However, on the basis of stability, QDs are superior to conventional HTMs. In terms of future PCE improvements, there is a huge upside for QDs.

## 6. Perovskite Quantum Dots as Light-Absorbing Layers

Perovskite quantum dots (PQDs) have been fabricated by researchers as light-absorbing layers due to their high photoluminescence quantum yield (PLQY) and tunable forbidden bands, which are beneficial for enhancing the light capture capability of the devices. Table 6 lists the details of the devices [128,129,130,131,132,133,134,135,136,137,138,139,140,141,142,143,144,145,146,147,148,149].

Among many PQDs, the most representative are the CsPbI_3_ QDs. In 2017, Sanehira et al. [146] innovatively proposed A-site cation halide salt treatments to improve the coupling between PQDs and doubling the film mobility, enabling increased photocurrent and achieving a PCE of 13.40%. The first planar heterojunction PSCs were prepared by Yuan et al. [149] in 2018 using undoped conjugated polymer poly [4,8-bis(2-ethylhexyl)oxy]benzo [1,2-b:4,5-b’]dithiophene-2,6-diyl-alt-3-fluoro-2-[(2-ethylhexyl)carbonyl]thieno [3,4-b]thiophene-4,6-diyl] (PTB7) and CsPbI_3_ QDs together, with the structure shown in Figure 10a. The combined effect of the PTB7 and CsPbI_3_ QDs facilitated the extraction of holes, and the final device showed an extremely low V_oc_ loss of just 0.42 eV, the lowest loss ever recorded at that moment. The final PCE was 12.55%. To further optimize the films of PQDs and improve the performance of the device, in 2019, Shi et al. [133] doped the rare-earth element Yb in situ into CsPbI_3_ quantum dot (Yb-CsPbI_3_ QD) films (Figure 10b), and the PLQY of the QDs was increased to about 86%. Additionally, defects on the surface of the CsPbI_3_ QD films were passivated by Yb^3+^ ions, reducing the non-radiative recombination of carriers and increasing the decay time from 44.3 ns to 51.4 ns. Ultimately, the best performance was achieved by the 20% Yb-based device at 13.12%. This work proved that the in situ doping of rare-earth elements was an effective strategy, but there are few reports on the in situ doping of rare-earth elements now.

The mixing ratio of ions also plays a significant role in the films’ PQDs, and ion engineering plays a key role in optimizing the QD films. Hao et al. [145] reduced the defects of the film by adjusting the ratio of FA^+^ ions to Cs^+^ ions and improved the electron and hole transport rates, ultimately achieving a PCE of 16.60% with a Cs_0.5_FA_0.5_PbI_3_ QD-based device. Li et al. [142] mixed narrow bandgap FAPbI_3_ QDs with wide bandgap CsPbI_3_ QDs to form a gradient heterojunction. By the proportional modulation of the hybrid QDs, a suitable energy band alignment was obtained, facilitating charge coupling, and the final PCE was 15.6%. In addition, the environmental stability was improved, as shown in Figure 10c.

PQDs generally require ligand exchange and the long-chain groups on their surfaces, which are not conducive to charge transport in the device. In 2019, Ling et al. [131] used short-chain phenylethylammonium (PEA) ligands instead of OAm ligands. The measurements revealed enhanced electronic coupling and improved hydrophobicity, with the final device achieving a 14.1% PCE. In 2022, Han et al. [144] used 1-propyl-3-methylimidazolium iodide ((Pmim)I) ionic liquid to replace the OA and OAm long-chain insulating ligands on the surface of QDs. The charge transfer rate of the film was increased (the mechanism is shown in Figure 10d). In addition, the ionic liquid significantly passivated the surface defects and the root-mean-square roughness decreased from 4.65 to 3.97 nm, resulting in a high-quality film with a PCE of 14.14%.

Sn-based PQDs have also been investigated. In 2017, Liu et al. [140] prepared CsSn_1−x_Pb_x_I_3_ QDs by mixing CsPbI_3_ QDs with CsSnI_3_ QDs. The hybrid QDs maintained a stable phase for several months and increased the speed of photogenerated electron transport with a final PCE of 0.1%. Following this, Wang et al. [143] synthesized CsSnI_3_ QDs via a single pot. Due to the addition of triphenyl phosphite (TPPi), the oxidation resistance of the PSC was substantially improved, maintaining a stable phase at room temperature for more than 90 days, and the PCE of the optimal device reached 5.03%. The addition of antioxidants made the CsSnI_3_ QDs films more stable and provided a simple method for producing highly stable lead-free inorganic PQDs, giving great potential for the widespread use of lead-free inorganic PSCs.

The PCE of quantum dot perovskite solar cells is currently improving, and has already increased to over 15% (Figure 10e). In the future, QDs as the light-absorbing layer will certainly achieve even greater breakthroughs.

**Figure 10 ijms-23-09482-f010:**
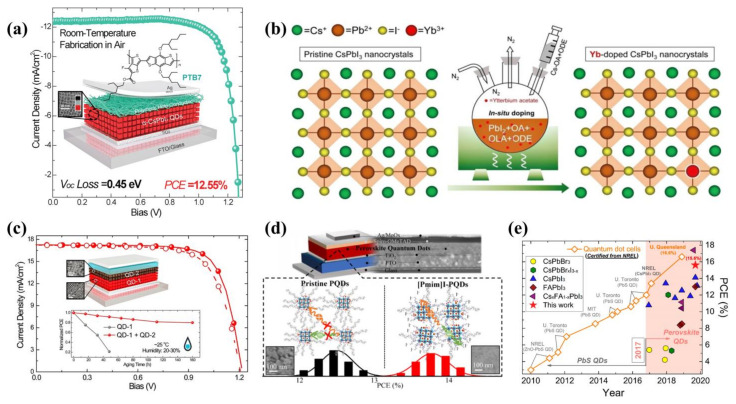
(**a**) Structure of planar heterojunction PSC. Reproduced from [149], with permission from Elsevier, 2018. (**b**) Illustration of the CsPbI_3_ QDs synthesis through in situ Yb doping. Reproduced from [133], with permission from the Royal Society of Chemistry, 2019. (**c**) Structure of hybrid PSC. Reproduced from [142], with permission from the American Chemical Society, 2019. (**d**) Mechanistic diagram of ligand action. Reproduced from [144], with permission from the American Chemical Society, 2022. (**e**) Progress of QD solar cells. Reproduced from [142], with permission from the American Chemical Society, 2019.

## 7. Quantum Dots as Luminescent Down-Shifting Materials

Currently, most PSCs can only use visible light in the 400–800 nm range, which is only about half of the solar radiation. Light in the under-ultraviolet (UV) and infrared (IR) regions cannot be absorbed by PSCs, resulting in an invisible waste of energy, so more and more researchers are looking to capture light in the UV and IR regions to make the most of it. In addition, UV light tends to cause the catalytic decomposition of materials such as TiO_2_, which can significantly reduce the stability of PSCs. Therefore, the use of UV and IR light can not only improve the PCE, but also the stability of the device. Table 7 lists the details of the devices [150,151,152,153,154,155,156,157,158].

In 2017, Wang et al. [150] deposited Mn-doped CsPbCl_3_ QDs on the transparent side of PSCs for the first time (Figure 11a). With a PLQY of 60% and a large Stokes shift (>200 nm), the effective use of radiation from 300 nm to 400 nm in the UV region increased the PCE from 17.97% to 18.57%. However, experiments showed that CsPbCl_3_:Mn luminescent down-shifting (LDS) film automatically degraded under moist heat or high temperature, which was detrimental to the stability. This work broke away from the traditional thinking of optimizing devices and opened up a new avenue for device optimization. In another study, Wang et al. [151] prepared broad-banded ZnSe QDs and used them as an LDS layer for PSCs in 2017 (Figure 11b). The highest J_sc_ were found to be 20.3 mA cm^−2^ and PCE was 17.3%. Respectively, when the QD concentration was 0.2 mg/L, in 2019, Bian et al. [153] prepared N-GQDs with a PLQY of 80%. Compared to CsPbCl_3_:Mn QDs, N-GQDs have stable chemical bonds and still work stably under humid and hot environments (Figure 11c). In addition, it was found that when the number of deposited N-GQDs reached three layers, the film was able to convert the harmful UV light into visible light with maximum efficiency. The final N-GQD-based device achieved a PCE of 16.02%.

CQDs have shown excellent performance in both ETL and HTL, but applications in the LDS layer have rarely been reported. In 2020, Maxim et al. [155] prepared sandwich-structured PSCs by dispersing CQDs in polymethyl methacrylate (PMMA) film and chlorobenzene, and then deposited the composite film on the outer side of the PSC. Owing to the down-conversion effect of CQDs, UV light was converted into visible light with a PCE of 17.86%. Following this, they used water as a dispersion medium for CQDs, avoiding the use of toxic chlorobenzene [158]. CQDs embedded in H_2_O (CQDs@H_2_O) resulted in a device with improved J_sc_ and FF compared to CQDs embedded in PMMA (CQDs@PMMA) films (Figure 11d). This work is the only report of CQDs as an LDS layer so far.

**Figure 11 ijms-23-09482-f011:**
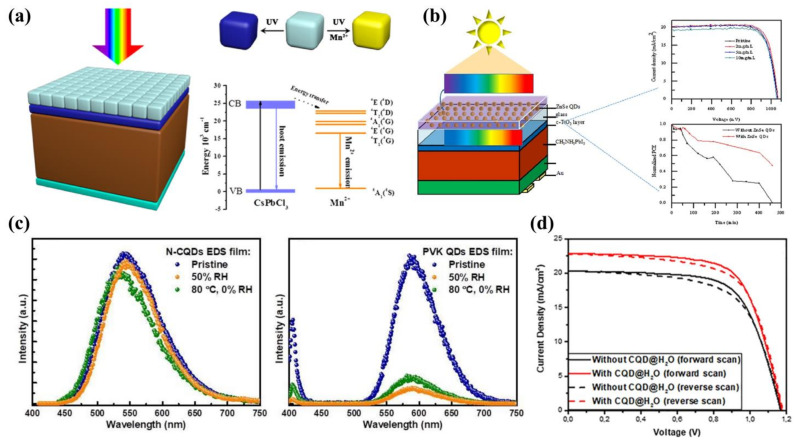
(**a**) Structural and energy band alignment diagrams based on CsPbI_3_:0.1Mn QD-based PSCs. Reproduced from [150], with permission from the American Chemical Society, 2017. (**b**) ZnSe QDs as the luminescent down-shifting layer. Reproduced from [151], with permission from Elsevier, 2017. (**c**) Stability of N-CQD LDS film and CsPbI_3_:Mn QD LDS film in the environment. Reproduced from [153], with permission from the Royal Society of Chemistry, 2019. (**d**) Forward and reverse J-V characteristics of different PSCs. Reproduced from [158], with permission from Elsevier, 2021.

In another study, Anizelli et al. [159] reported that luminescent down-shifting QDs enable the filtering of UV radiation with increased solar cell stability. The PCE of the non-encapsulated device with the application of a luminescent down-shifting layer dropped by ~18% over 30 h, which was compared to ~97% for an unfiltered device, also without encapsulation. In addition, Guo et al. [160] reported that photoluminescent materials can be directly added to monitor the performance of PSCs. They found that photoluminescent spectroscopy was a more sensitive method than UV visible light absorption for characterizing the initial stages of perovskite degradation. Mahon et al. [161] designed the useful technique of tracing PL kinetics under continuous illumination at the “seconds-to-minutes” timescale, which was able to be applied for the analysis of PSCs at various stages of their fabrication and lifespan.

The phototransfer layer effectively converts UV to visible light, improving PCE and stability. The photoluminescent materials are able to monitor the performance of PSCs and add a pathway for QDs to optimize PSCs.

## 8. Summary and Outlook

Although the introduction of QDs can increase the performance and stability of PSCs by a substantial margin, there are some problems with QDs that prevent them from unlocking their full potential, and some aspects need to be improved.

First of all, the manufacturing method of QDs is a key step. The predominant fabrication methods are currently thermal injection and low-temperature solution. Not only are these methods required to introduce the ligands, but also require high temperatures for thermal injection, which can have a negative effect on the perovskite film and increase the manufacturing costs. Therefore, it is extremely important to help to develop new methods of preparation. In this paper, many excellent methods are mentioned, such as microwave synthesis for the uniform heating of QDs, thus controlling the crystallization rate. The supersaturated recrystallisation method allows the preparation of QDs with a high degree of stability. In addition, pulsed laser radiation is the latest method for the preparation of QDs, which is rapid. Most importantly, the crystallization of the film is not affected by the introduction of ligands during the preparation process. These methods have great potential for optimizing QDs and increasing yields.

Secondly, the element doping of QDs is also an effective tactic to increase the PCE. Element doping can improve the rate of charge transport, but not all element doping will improve the performance of PSCs. Some ions do not match the lattice of perovskite, which will lead to a reduction in device performance. For this reason, we need to find matching QDs for the study of this approach. Sn^2+^ ions are considered to be an effective alternative to the toxic Pb^2+^ ions in perovskite. Future research on Sn-based PSCs should be increased.

Thirdly, ligand engineering is also an effective method for modifying QDs. Ligands can be divided into two types: long-chain ligands and short-chain ligands. Perovskite QDs are generally stabilized by long-chain insulating ligands, but the long-chain ligands inhibit charge transport. Short-chain ligands, although beneficial for charge transport, can cause the agglomeration of QDs. It is necessary to find suitable ligands to exchange with the oleic acid ligands on the surface of the perovskite to improve the conductivity, while maintaining the black stable phase of the perovskite. There have been attempts by researchers to prepare ligand-free perovskite with high charge transport, but with a PCE below 10%. This method is also a promising strategy.

Fourthly, the choice of solvent is also important in the production of QDs. Highly polar solvents break the ionic bonds formed between the long-chain ligands and the perovskite surface, while highly non-polar solvents are unable to remove the long-chain ligands, and the QDs will agglomerate. Therefore, the use of solvents with medium polarity is an effective route. Most of the moderately polar solvents used today are low-toxicity green solvents, such as ethyl acetate and methyl acetate, which can remove long-chain ligands while maintaining the stability of QDs.

Fifthly, since many PSCs contain toxic Pb, a sustainable procedure to recycle the cells after their operational lifetime is required to prevent the exposure of Pb to the environment. In 2016, Binek et al. [162] demonstrated an environmentally responsible and cost-efficient recycling process for MAPbI_3_-based PSCs. The toxic PbI_2_ can be recycled, and after recrystallization, can be employed to prepare the device, with a PCE reaching 13.5%. In addition, they were able to recycle the expensive FTO/glass substrates several times. In 2017, Xu et al. [163] developed the in situ recycling of PbI_2_ from thermal decomposition CH_3_NH_3_PbI_3_ perovskite films for efficient PSCs. This approach can reduce the risk of Pb outflow. In 2022, Hong et al. [164] investigated the novel synthesis of whitlockite (WH,Ca_18_Mg_2_(HPO_4_)_2_(PO_4_)_12_), which can act as a high-performance Pb^2+^ absorbent. Their method provided an efficient way to remove Pb^2+^ ions from water and recycle the PSCs. We should explore more novel approaches to recycling the Pb in PSCs, as this will bring us one step closer to commercialization.

All of these approaches can be used to optimize the performance of QDs and strengthen the performance of the PSCs. We are confident that QDs will make a big difference in the field of PSCs.

## Figures and Tables

**Figure 1 ijms-23-09482-f001:**
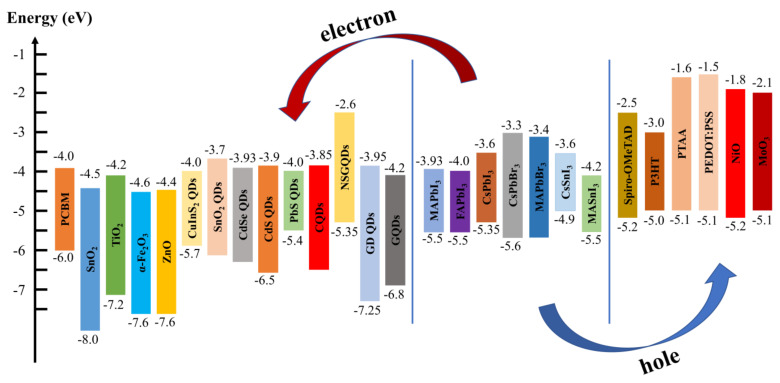
Energy level diagrams of various QDs applied in PSCs.

**Figure 7 ijms-23-09482-f007:**
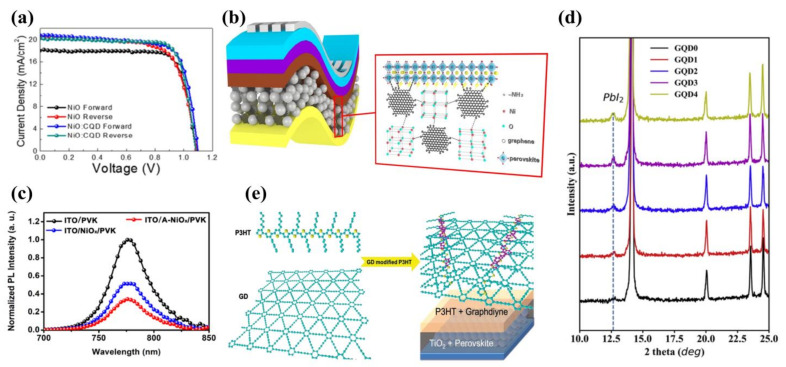
(**a**) J-V curves for forward and reverse scanning. Reproduced from [79], with permission from Elsevier, 2019. (**b**) AGQDs optimize NiO_x_ [81]. (**c**) Steady-state photoluminescence spectra. Reproduced from [81], with permission from the American Chemical Society, 2020. (**d**) XRD pattern shows the presence of PbI_2_ diffraction peaks. Reproduced from [80], with permission from Elsevier, 2019. (**e**) P3HT modified with graphdiyne. Reproduced from [83], with permission from Wiley, 2015.

**Table 1 ijms-23-09482-t001:** Details of QDs as additives in ETL.

QDs	Device Structure	V_oc_	J_sc_	FF %	PCE %	Ref.
CQDs (10 nm)	ITO/TiO_2_:CQDs/MAPbI_3_Cl_3−x_/Spiro-OMeTAD/Au	1.136	21.36	78	18.89	[26]
CQDs (4 nm)	FTO/PEDOT:PSS/MAPbI_3_/PCBM:CQDs/BCP/Ag	0.97	22.30	79.6	18.10	[27]
CQDs (~4.8 nm)	FTO/c-TiO_2_/m-TiO_2_:CQDs/MAPbCl_x_I_3−x_/Spiro-OMeTAD/Au	1.019	22.64	71.6	16.40	[25]
g-C_3_N_4_ QDs (5~10 nm)	ITO/SnO_2_:g-C_3_N_4_ QDs/CsFAMA/Spiro-OMeTAD/Au	1.176	24.03	78.3	22.13	[28]
Red CQDs (~)	ITO/SnO_2_:RCQs/Cs_0.05_FA_0.81_MA_0.14_PbI_2.25_Br_0.45_/Spiro-OMeTAD /MoO_3_/Au	1.14	24.1	82.9	22.77	[29]
GQDs (~5 nm)	FTO/Au/SnO_2_:GQDs/ZnO/Perovskite/Spiro-OMeTAD/Au	1.172	22.85	74	19.81	[37]
GQDs (5~10 nm)	ITO/SnO_2_:GQDs/MAPbI_3_/Spiro-OMeTAD/Au	1.13	23.05	78	20.31	[30]
GQDs (~)	FTO/SnO_2_:GQDs/CsFAMA/Spiro-OMeTAD/Ag	1.10	21.62	78	18.55	[36]
GQDs (6 nm)	ITO/c-TiO_2_/m-TiO_2_:GQDs/Cs_0.05_(MA_0.17_FA_0.83_)_0.95_Pb(I_0.83_Br_0.17_)_3_/Spiro-OMeTAD/Au	0.97	21.92	67	14.36	[33]
GQDs (~5 nm)	ITO/PCBM:GQDs/MAPbI_3_/Spiro-OMeTAD/Au	1.09	22.03	73	17.56	[31]
GQDs (~)	ITO/SnO_2_:GQDs/MAFAPbI_x_Cl_3−x_ /Spiro-OMeTAD/Ag	1.11	24.40	78	21.10	[34]
GQDs (3~5 nm)	FTO/c-TiO_2_/m-TiO_2_:GQDs/Perovskite/Spiro-OMeTAD/Ag	1.08	24.92	76	20.45	[32]
GQDs (5.4~10.9 nm)	FTO/SnO_2_:GQDs/CsFAMA/Spiro-OMeTAD/Au	1.08	23.5	77	19.6	[35]
CdSe QDs (10 nm)	ITO/PEDOT:PSS/CH_3_NH_3_PbI_3−x_Cl_x_/PCBM:CdSe QDs/LiF/Ag	0.90	20.96	73.16	13.73	[38]
CdS QDs (4~5 nm)	FTO/TiO_2_:CdS QDs/CH_3_NH_3_PbI_3_/Spiro-OMeTAD/Au	0.94	16.7	64	10.52	[44]
PbS QDs (~5 nm)	FTO/TNTs:PbS QDs/Cs_0.05_(FA_0.85_MA_0.15_)_0.95_Pb(I_0.85_Br_0.15_)_3_/Spiro-OMeTAD/Au	1.14	23.38	56.03	14.95	[39]
PbS QDs (1~3 nm)	ITO/ZnO:PbS QDs-TBAI-80/MAPbI_3_/PCBM/Ag	1.14	22.80	78.99	20.53	[42]
BP QDs (2~8 nm)	Glass/FTO/SnO_2_:BP QDs/BP QDs@A-CsFAMA/Spiro-OMeTAD/Ag	1.22	23.53	79.6	22.85	[40]
BP QDs (~)	FTO/SnO_2_:BP QDs/(FAPbI_3_)_0.97_(MAPbBr_3_)_0.03_/Spiro-OMeTAD/Ag	1.13	24.4	76.1	21.0	[41]
SnO_2_ QDs (4~5 nm)	FTO/c-TiO_2_/m-TiO_2_:SnO_2_ QDs/MAPbI_3_/Spiro-OMeTAD/Ag	1.13	22.36	75.67	19.09	[43]

**Table 2 ijms-23-09482-t002:** Details of QDs as additives in perovskite films.

QDs	Device Structure	V_oc_	J_sc_	FF %	PCE %	Ref.
CQDs (3~5 nm)	FTO/TiO_2_/ZrO_2_/CH_3_NH_3_PbI_3_:CQDs/Carbon	0.79	16.40	61.2	7.62	[50]
CQDs (~)	ITO/PTAA/CH_3_NH_3_PbI_3_:CQDs/Ti/Cu	1.101	23.13	75.28	19.17	[53]
CQDs (1.62~7.36 nm)	ITO/NiO/CH_3_NH_3_PbI_3_:CQDs/PCB_61_M/BCP/Ag	1.07	21.68	78.78	18.24	[52]
CQDs (3~5 nm)	ITO/c-TiO_2_/m-TiO_2_/MAPbI_3_:CQDs/Spiro-OMeTAD/Ag	1.13	22.19	75	18.81	[16]
EACQDs (2.65~5.37 nm)	FTO/c-TiO_2_/m-TiO_2_/ZrO_2_/MAPbI_3_:EACQDs/Carbon	1.021	22.72	65.29	15.14	[56]
ASCQDs (3.43 nm)	FTO/c-TiO_2_/m-TiO_2_/ZrO_2_/CH_3_NH_3_PbI_3_:ASCQDs/Carbon	1.03	23.84	61.13	14.95	[55]
NCDs (12 nm)	FTO/TiO_2_/CH_3_NH_3_PbI_3_:NCDs/Spiro-OMeTAD/Ag	1.08	20.66	73.91	16.49	[51]
A-CQDs (4 nm)	FTO/c-TiO_2_/m-TiO_2_/ZrO_2_/MAPbI_3_:A-CQDs/Carbon	0.94	22.35	51.27	13.28	[54]
GQDs (~3 nm)	FTO/c-TiO_2_/MAPbI_3_:GQDs/Spiro-OMeTAD/Au	1.14	24.17	71.6	19.70	[58]
GQDs (~20 nm)	FTO/c-TiO_2_/MAPbI_3_:GQDs/Spiro-OMeTAD/Au	1.04	22.58	75	18.34	[57]
GN-GQDs (1.5~5.5 nm)	FTO/NiO/MAPbI_3_:GN-GQDs/PCBM/BCP/Ag	1.06	23.40	80	19.8	[59]
F-GQDs (2~8 nm)	PEN/ITO/SnO_2_/Rb_0.05_Cs_0.05_(FA_0.83_MA_0.17_)_0.90_ Pb(I_0.95_Br_0.05_)_3_:F-GQDs/Spiro-OMeTAD/Ag	1.106	22.87	80.67	20.40	[60]
BP QDs (3~5 nm)	ITO/SnO_2_/CsPbI_2_Br:BP QDs/Spiro-OMeTAD/Au	1.25	15.86	78	15.47	[67]
Si QDs (~)	FTO/c-TiO_2_/m-TiO_2_/CsPbBr_3_:Si QDs/Spiro-OMeTAD/Ag	1.42	7.80	75	8.31	[66]
AgI QDs (~)	ITO/MAPbI_3_:AgI QDs/PCBM/Ag	1.01	22.94	70.83	16.41	[63]
FeOOH QDs (~3 nm)	FTO/c-TiO_2_/m-TiO_2_/Cs_0.05_FA_0.81_MA_0.14_Pb Br_0.45_I_2.55_:FeOOH QDs/Spiro-OMeTAD/Au	1.17	23.12	77.6	21.0	[64]
PbS QDs (~3.5 nm)	ITO/c-TiO_2_/m-TiO_2_/MAPbI_3_:PbS QDs/Spiro-OMeTAD/Au	1.08	22.50	75	18.22	[65]
PbS QDs (~4 nm)	ITO/c-TiO_2_/m-TiO_2_/MAPbI_3_:PbS QDs/Spiro-OMeTAD/Au	1.02	23.50	77.2	18.60	[62]
PbS QDs (4.7 nm)	FTO/SnO_2_/FAPbI_3_:PbS QDs/Spiro-OMeTAD/Au	1.06	21.80	75	18.00	[68]
PbS QDs (5~7 nm)	ITO/PEDOT: PSS/CsPb_0.5_Sn_0.5_BrI_2_: PbS QDs/PCBM/Ag	0.66	20.08	61	8.03	[73]
rGO-Sn QDs (6 nm)	FTO/SnO_2_/Al_2_O_3_/Graphene/FA_0.8_MA_0.2_SnI_3_:Sn QDs/Spiro-OMeTAD/Au	0.68	16.63	68.1	7.70	[71]
SnS QDs (3~4 nm)	FTO/c-TiO_2_/CH_3_NH_3_PbI_3_:SnS QDs/Spiro-OMeTAD/Au	1.036	22.70	71.6	16.80	[61]
SnS QDs (~10 nm)	FTO/m-TiO_2_/Cs_2_AgBiBr_6_:SnS QDs/Carbon	1.02	3.74	51	1.95	[75]
WS_2_ QDs (3 nm)	FTO/c-TiO_2_/m-TiO_2_/ZrO_2_/MAPbI_3_:WS_2_ QDs/Carbon	1.057	23.08	69.09	16.85	[69]
CdSe QDs (3.8 nm)	FTO/c-TiO_2_/m-TiO_2_/CsPbBrI_2_:MPA-CdSe QDs/Carbon	1.25	14.47	80.1	14.49	[72]
Ti_3_C_2_Cl_x_ QDs (8 nm)	FTO/SnO_2_/Rb_0.05_Cs_0.05_(FA_0.83_MA_0.17_)_0.90_ Pb(I_0.83_Br_0.17_)_3_:T_3_C_2_Cl_x_ QDs/PCBM/Ag	1.19	22.27	80.42	21.31	[70]
CuInSe_2_ QDs (12~25 nm)	ITO/SnO_2_/MAPbI_3_:CuInSe_2_ QDs/Spiro-OMeTAD/Ag	1.09	21.09	78	18.04	[74]

**Table 3 ijms-23-09482-t003:** Details of QDs as additives in HTL.

QDs	Device Structure	V_oc_	J_sc_	FF %	PCE %	Ref.
CQDs (4 nm)	FTO/SnO_2_/(FAPbI_3_)_0.95_(MAPbBr_3_)_0.05_/Spiro-OMeTAD:GQDs/Ag	1.06	24.17	79.41	20.41	[77]
CQDs (4~6 nm)	ITO/GO:CQDs/CH_3_NH_3_PbI_3_/PCBM/BCP/Ag	0.953	21.0	80.1	16.2	[78]
CQDs (3.2 nm)	ITO/NiO:CQDs/CH_3_NH_3_PbI_3_/PCBM/BCP/Ag	1.08	20.22	77.15	16.91	[79]
GQDs (15 nm)	FTO/PEDOT:PSS/GQDs/CH_3_NH_3_PbI_3_/PCBM/BCP/Ag	1.002	21.41	75.31	16.16	[80]
AGQDs (2 nm)	ITO/NiO:AGQDs/(FA_0.83_MA_0.17_)_0.95_Cs_0.05_Pb(I_0.9_Br_0.1_)_3_/PCBM/BCP/Ag	1.067	22.5	81.5	19.55	[81]
Graphdiyne QDs (3~5 nm)	FTO/TiO_2_/GD QDs/CH_3_NH_3_PbI_3_:GD QDs/Spiro-OMeTAD:GD QDs/Au	1.124	22.48	78.7	19.89	[82]
Graphdiyne QDs (50 nm)	FTO/TiO_2_/CH_3_NH_3_PbI_3_/P3HT:GD QDs/Au	0.941	21.7	71.3	14.58	[83]
PbSO_4_(PbO)_4_ QDs (5 nm)	ITO/SnO2/CsFAMA/Spiro-OMeTAD:PbSO_4_(PbO)_4_ QDs/Au	1.142	24.80	80	22.66	[84]

**Table 4 ijms-23-09482-t004:** Details of QDs as ETM.

QDs	Device Structure	V_oc_	J_sc_	FF %	PCE %	Ref.
TiO_2_ QDs (3.6 nm)	FTO/TiO_2_ QDs/m-TiO_2_/CH_3_NH_3_PbI_3_/Spiro-OMeTAD/Au	1.063	22.48	71	16.97	[89]
ZnO QDs (~)	ITO-PET/Graphene/ZnO QDs (Apjet)/CH_3_NH_3_PbI_3_/Spiro-OMeTAD/Ag	0.935	16.80	62	9.73	[88]
ZnO/rGO QDs (5 nm)	FTO/ZnO/rGO QDs/CH_3_NH_3_PbI_3_/Spiro-OMeTAD/Au	1.03	21.7	68	15.2	[90]
SnO_2_ QDs (1.7~3.3 nm)	ITO/SnO_2_ QDs/MAPbI_3_/Spiro-OMeTAD/Ag	1.08	21.85	74.28	17.66	[98]
SnO_2_ QDs (2~4 nm)	FTO/SnO_2_ QDs/MA_0.7_FA_0.3_PbI_3_/Spiro-OMeTAD/Au	1.08	23.40	74	18.80	[95]
SnO_2_ QDs (1.7~3.3 nm)	FTO/Al:SnO_2_ QDs/MAPbI_3_/Spiro-OMeTAD/Ag	1.06	22.78	75.41	18.20	[96]
SnO_2_ QDs (3~5 nm)	FTO/SnO_2_ QDs/Cs_0.05_(MA_0.17_FA_0.83_)_0.95_Pb(I_0.83_Br_0.17_)_3_/Spiro-OMeTAD/Ag	1.13	23.05	79.8	20.79	[94]
SnO_2_ QDs (2.9 nm)	ITO/SnO_2_ QDs/MAPbI_3_/Li-doped Spiro-OMeTAD/Au	1.12	21.61	77	18.71	[93]
SnO_2_ QDs (3 nm)	ITO/SnO_2_ QDs/Cs_0.05_FA_0.81_MA_0.14_PbI_2.25_Br_0.45_/Spiro-OMeTAD/Carbon	1.08	22.19	56.64	13.64	[97]
BP QDs (4.7 nm)	ITO-PEN/BP QDs/FA_0.85_MA_0.15_PbI_0.25_Br_0.5_/Spiro-OMeTAD/Au	1.03	16.77	65.2	11.26	[92]
CdSe QDs (5.5 nm)	ITO/PEDOT:PSS/MAPbI_3_/CdSe QDs/LiF/Ag	0.99	20.5	69.9	14.2	[91]

**Table 5 ijms-23-09482-t005:** Details of QDs as HTM.

QDs	Device Structure	V_oc_	J_sc_	FF %	PCE %	Ref.
CQDs (~)	FTO/c-TiO_2_/m-TiO_2_/MAPbI_3_/CQDs/Au	0.515	7.83	74	3	[110]
PbS QDs (3 nm)	ITO/PbS QDs/MAPbI_3_/PCBM/Al	0.86	12.1	72	7.5	[105]
PbS QDs (3.6 nm)	FTO/c-TiO_2_/m-TiO_2_/CH_3_NH_3_PbI_3_/PbS QDs/Au	0.87	18.69	49	7.88	[106]
PbS QDs (3.5 nm)	FTO/c-TiO_2_/m-TiO_2_/CH_3_NH_3_PbI_3_/PbS QDs/Au	0.97	19.03	61.34	11.32	[126]
PbS QDs (~)	FTO/TiO_2_/m-TiO_2_/CH_3_NH_3_PbI_3_/PbS QDs/Au	0.8	29.3	83	19.52	[125]
CuInS_2_ QDs (4 nm)	FTO/c-TiO_2_/m-TiO_2_/MAPbI_3_/CuInS_2_/ZnS QDs/Au	0.924	18.6	48.7	8.38	[107]
CuIn_1.5_Se_3_ QDs (4 nm)	ITO/SnO_2_/MAPbBr_3_/CuIn_1.5_Se_3_ QDs/Au	0.979	20.46	68.5	13.72	[113]
CuInSe_2_ QDs (7 nm)	ITO/SnO_2_/FAMAPbI_3_BrCl/CuInSe_2_ QDs/Au	0.86	22.5	66	12.8	[118]
SnS QDs (4~7 nm)	FTO/c-TiO_2_/(CsPbI_3_)_0.05_(FAPbI_3_)_0.79_(MAPbI_3_)_0.16_/SnS QDs/Au	0.944	22.96	63.3	13.72	[121]
MoS_2_ QDs (4~7 nm)	FTO/c-TiO_2_/m-TiO_2_/CsPbBr_3_/MoS_2_ QDs/Carbon	1.307	6.55	79.4	6.80	[116]
Cu_2_O QDs (8~10 nm)	FTO/c-TiO_2_/m-TiO_2_/Cs_0.05_FA_0.81_MA_0.14_PbI_2.55_Br_0.45_/Cu_2_O QDs/Au	1.15	22.2	74.2	18.90	[117]
CsSnBr_3_ QDs (20 nm)	FTO/SnO_2_ QDs/CsPbBr_3_/CsSnBr_3_ QDs/Carbon	1.61	7.8	84.4	10.60	[119]
Ag-In-Ga-S QDs (~)	FTO/c-TiO_2_/m-TiO_2_/CsPbBr_3_/AIGS QDs/Carbon	1.46	7.43	80.31	8.46	[120]
Cu_12_Sb_4_S_13_ QDs (5.7 nm)	FTO/c-TiO_2_/CH_3_NH_3_PbI_3_/Cu_12_Sb_4_S_13_ QDs/Au	1.05	21.85	61.6	14.13	[122]
Cu_12_Sb_4_S_13_ QDS (5~6 nm)	FTO/c-TiO_2_/m-TiO_2_/CsPbI_3_ QDs/Cu_12_Sb_4_S_13_ QDs/Au	1.04	18.28	52.9	10.02	[124]
Cu_12_Sb_4_S_13_ QDs (9 nm)	FTO/c-TiO_2_/CH_3_NH_3_PbI_3_/Cu_12_Sb_4_S_13_ QDs/Au	0.8	18.08	45	6.5	[114]
Cu_2_ZnSnS_4_ QDs (20 nm)	FTO/c-TiO_2_/CH_3_NH_3_PbI_3_/Cu_2_ZnSnS_4_ QDs/Au	1.06	20.54	58.7	12.75	[108]
Cu_2_ZnSnS_4_ QDs (8 nm)	FTO/m-TiO_2_/c-TiO_2_/CsPbBr_3_/Cu_2_ZnSnS_4_ QDs/Ag	0.94	7.36	70.01	4.84	[123]
Cu_2_ZnSnS_4_ QDs (5.15 nm)	ITO/Cu_2_ZnSnS_4_-LF QDs/Perovskite/PCBM/Ag	0.92	20.7	81	15.4	[109]
Cu_2_ZnSnSe_4_ QDs (~)	FTO/TiO_2_/CH_3_NH_3_PbI_3_/Cu_2_ZnSnSe_4_ QDs/Au	0.808	19.37	62.1	9.72	[111]
CuIn_0.1_Ga_0.9_(S_0.9_Se_0.1_)_2_ QDs (10 nm)	FTO/c-TiO_2_/m-TiO_2_/CH_3_NH_3_PbI_3_/CIGSSe QDs/Au	0.94	17.66	54.88	9.15	[112]
CsSnBr_2_I QDs (1.7–3.8 nm)	FTO/c-TiO_2_/m-TiO_2_/CsPbBr_3_/CsSnBr_2_I QDs/Carbon	1.39	8.70	76	9.13	[115]
CuInS_2_@ZnS QDs (5.6~6.4 nm)	FTO/c-TiO_2_/m-TiO_2_/CsPbBr_3_/CuInS_2_@ZnS QDs/Carbon	1.45	7.47	77.73	8.42	[103]
CdZnSe@ZnSe QDs (6 nm)	FTO/c-TiO_2_/m-TiO_2_/CsPbBr_3_/CdZnSe@ZnSe QDs/Carbon	1.498	7.25	79.6	8.65	[104]

**Table 6 ijms-23-09482-t006:** Details of PQDs as light-absorbing layers.

QDs	Device Structure	V_oc_	J_sc_	FF %	PCE %	Ref.
MAPbI_3_ QDs (2~3 nm)	FTO/c-TiO_2_/m-TiO_2_/MAPbI_3_ QDs/Iodine redox electrolyte/Pt	0.71	15.82	59	6.54	[137]
MAPbBr_3_ QDs (2 nm)	FTO/c-TiO_2_/m-TiO_2_/MAPbBr_3_ QDs/PTAA/Au	1.11	14.07	73	11.40	[138]
FAPbI_3_ QDs (17.7 nm)	ITO/SnO_2_/FAPbI_3_ QDs/Spiro-OMeTAD/Au	1.10	11.83	64	8.38	[148]
FAPbI_3_ QDs (7~12 nm)	ITO/c-TiO_2_/FAPbI_3_ QDs/Spiro-OMeTAD/Au	1.05	13.3	67	9.40	[139]
CsPbI_3_ QDs (3~12.5 nm)	ITO/c-TiO_2_/CsPbI_3_ QDs/Spiro-OMeTAD/MoO_x_/Al	1.23	13.47	65	10.77	[147]
CsPbI_3_ QDs (10 nm)	FTO/c-TiO_2_/CsPbI_3_ QDs/MoO_x_/Al	1.20	14.37	78	13.40	[146]
CsPbI_3_ QDs (8 nm)	FTO/c-TiO_2_/CsPbI_3_ QDs/PTB7/MoO_3_/Ag	1.27	12.39	80	12.55	[149]
CsPbI_3_ QDs (10 nm)	FTO/c-TiO_2_/CsPbI_3_ QDs/Spiro-OMeTAD/Au	1.04	16.98	67	11.87	[130]
Yb-CsPbI_3_ QDs (12 nm)	FTO/c-TiO_2_/Yb-CsPbI_3_ QDs/PTB7/MoO_3_/Ag	1.25	14.18	74	13.12	[133]
CsPbI_3_ QDs (12 nm)	FTO/c-TiO_2_/CsPbI_3_ QDs/Spiro-OMeTAD/Au	1.11	14.80	74	12.15	[132]
CsPbI_3_ QDs (~)	FTO/c-TiO_2_/CsPbI_3_ QDs/PTAA/MoO_3_/Ag	1.25	14.96	76	14.10	[131]
CsPbI_3_ QDs (9.1 nm)	FTO/c-TiO_2_/m-TiO_2_/CsPbI_3_ QDs/Spiro-OMeTAD/Au	1.06	17.77	76	14.32	[134]
CsPbI_3_ QDs (30 nm)	FTO/c-TiO_2_/CsPbI_3_ QDs/Spiro-OMeTAD/Au	1.06	16.87	75	13.38	[135]
CsPbI_3_ QDs (~)	FTO/NiO_x_/CsPbI_3_ QDs/C_60_/ZnO/Ag	1.19	14.25	77.6	13.1	[136]
CsPbBr_3_ QDs (7.2 nm)	FTO/c-TiO_2_/m-TiO_2_/CsPbBr_3_ QDs/Spiro-OMeTAD/Au	0.86	8.55	57	4.21	[128]
CsPbBr_3_ QDs (15~20 nm)	FTO/ZnO/CsPbBr_3_ QDs/Spiro-OMeTAD/Au	1.43	6.17	77	6.81	[129]
CsSn_0.6_Pb_0.4_I_3_ QDs (11~14 nm)	FTO/c-TiO_2_/m-TiO_2_/CsSn_0.6_Pb_0.4_I_3_ QDs/Spiro-OMeTAD/Au	0.26	0.90	42	0.1	[140]
CsSnI_3_ QDs (30 nm)	FTO/PEDOT:PSS/CsSnI_3_ QDs/PCBM/Ag	0.42	23.79	41	4.13	[143]
Cs_0.5_FA_0.5_PbI_3_ QDs (12 nm)	ITO/SnO_2_/Cs_0.5_FA_0.5_PbI_3_ QDs/Spiro-OMeTAD/Au	1.17	18.30	78	16.60	[145]
α-CsPbI_3_/FAPbI_3_ QDs (~)	FTO/TiO_2_/α-CsPbI_3_ QDs/FAPbI_3_ QDs/PTAA/MoO_3_/Ag	1.22	17.26	0.74	15.6	[142]
μGR-CsPbI_3_ QDs (10 nm)	FTO/TiO_2_/μGR-CsPbI_3_ QDs/PTAA/Au	1.18	13.59	72.6	11.64	[141]
[Pmim]I-CsPbI_3_ QDs (~)	FTO/TiO_2_/[Pmim]I-CsPbI_3_ QDs/Spiro-OMeTAD/MoO_x_/Ag	1.20	15.65	75.1	14.14	[144]

**Table 7 ijms-23-09482-t007:** Details of QDs as luminescent down-shifting layers.

QDs	Device Structure	Voc	Jsc	FF %	PCE %	Ref.
CQDs@PMMA QDs (3~7 nm)	CQDs@PMMA/Glass/FTO/SnO_2_/Cs_0.05_(MA_0.17_FA_0.83_)_0.95_Pb(I_0.84_Br_0.16_)_3_Spiro-OMeTAD/Au	1.13	23.21	68.19	17.86	[155]
CQDs@H_2_O QDs (3~7 nm)	CQDs@H_2_O/Glass/FTO/SnO_2_/Cs_0.05_(MA_0.17_FA_0.83_)_0.95_Pb(I_0.84_Br_0.16_)_3_/Spiro-OMeTAD/Au	1.18	22.8	65	17.60	[158]
N-GQDs (3.8 nm)	N-GQDs/Glass/FTO/TiO_2_/γ-CsPbI_3_/PTAA/Au	1.106	19.15	75.6	16.02	[153]
ZnSe QDs (~)	ZnSe QDs/Glass/FTO/c-TiO_2_/CH_3_NH_3_PbI_3_/Spiro-OMeTAD/Au	1.1	20.3	79.9	17.30	[151]
CdSe/CdS QDs (~)	CdSe/CdS QDs/Glass/ITO/NiO/CsFAPbI_3_/C_60_/BCP/Ag	1.143	23.6	76.8	20.70	[156]
CsPbCl_3_:Mn QDs (8 nm)	CsPbCl_3_:Mn QDs/Glass/FTO/TiO_2_/CH_3_NH_3_PbI_3_/Spiro-OMeTAD/Au	1.105	22.03	76.3	18.57	[150]
Ce^3+^-CsPbI_3_ QDs (~)	Ce^3+^-CsPbI_3_ QDs/Glass/FTO/c-SnO_2_/IBrFA_0.83_MA_0.17_Pb(I_0.83_Br_0.17_)_3_/PDPP4T/Au	1.19	24.13	79.7	22.16	[154]
CsPbBr_3_@SiO_2_ QDs (8 nm)	CsPbBr_3_@SiO_2_ QDs/Glass/FTO/c-TiO_2_/CH_3_NH_3_PbI_3_/Spiro-OMeTAD/Au	1.06	24.6	79.6	20.80	[152]
CsPbBr_3_ QDs (12 nm)	CsPbBr_3_ QDs/FTO/Ce-TiO_2_/CsFAMAPbI_3_/Spiro-OMeTAD/Au/Al_2_O_3_	1.112	22.26	78.13	19.34	[157]

## Data Availability

Not applicable.

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
