# Peer review of "Selection, Preparation and Application of Quantum Dots in Perovskite Solar Cells"

_ijms, 2022, doi:10.3390/ijms23169482_

Round 1

Reviewer 1 Report

The authors describe a review article entitled “Selection, preparation and application of quantum dots in perovskite solar cells”. The topic of the manuscript is interesting, and the manuscript constitutes an interesting review concerning the development of perovkistes for solar cells applications.

The work is well-written and a well-constructed introduction has been established by the authors. Sufficient spectra and figures are included in the manuscript for comprehension and clarity. Interesting and convincing results are also presented in this work. Overall, I think that this is a manuscript that I recommend for publication after inclusion of minor revisions.

1) In this review, a lot of abbreviations are used. Notably, in the different tables summarizing the performances of solar cells. Please define these different acronyms, perhaps in a list of abbreviations.

2) Quantum dots are made of toxic metals. This point is not evoked and could constitute a limitation for future developments. Please add a paragraph concerning this point.

3) What about recycling of these solar cells. Please comments.

Author Response

Responses to Reviewers (Manuscript ID: ijms-1863461)

To Reviewers:

Thank you very much for your pertinent comments and valuable suggestions as well as recommendation on our paper. These suggestions are very important for us to improve the quality of our manuscript. We have revised the manuscript according to them point by point as follows.

To Reviewer #1

Comments and Suggestions for Authors:

The authors describe a review article entitled “Selection, preparation and application of quantum dots in perovskite solar cells”. The topic of the manuscript is interesting, and the manuscript constitutes an interesting review concerning the development of perovskite for solar cells applications.

The work is well-written and a well-constructed introduction has been established by the authors. Sufficient spectra and figures are included in the manuscript for comprehension and clarity. Interesting and convincing results are also presented in this work. Overall, I think that this is a manuscript that I recommend for publication after inclusion of minor revisions.

  • In this review, a lot of abbreviations are used. Notably, in the different tables summarizing the performances of solar cells. Please define these different acronyms, perhaps in a list of abbreviations.

Author reply: Thank you very much! According to the valuable suggestion, we have defined these acronyms and made a list of abbreviations at the Table S1 in the Supplementary Material. (see below)

Table S1. A list of abbreviations in different tables

Abbreviation

Full name

CQDs

carbon quantum dots

GQDs

graphene quantum dots

RCQDs

red carbon quantum dots

g-C3N4 QDs

graphitized carbon nitride quantum dots

EACQDs

carbon quantum dots prepared in ethyl acetate solvent

ASCQDs

anti-solvent carbon quantum dots

NCDs

nitrogen doped carbon dots

A-CQDs

carbon quantum dots prepared from acetone

CA-CQDs

carbon quantum dots prepared from anhydrous citric acid

GN-GQDs

graphite−nitrogen doped graphene quantum dots

F-GQDs

F atoms doped in graphene quantum dots

AGQDs

graphene quantum dots with amino groups

N-GQDs

nitrogen-doped graphene quantum dots

CQDs@PMMA

polymethyl methacrylate with embedded carbon quantum dots

CQDs@H2O

H2O with embedded carbon quantum dots

CdSe QDs

CdSe quantum dots

CdS QDs

CdS quantum dots

PbS QDs

PbS quantum dots

BP QDs

black phosphorus quantum dots

SnO2 QDs

SnO2 quantum dots

rGO-Sn QDs

reduced graphene oxide sheets anchored with Sn quantum dots

AIGS QDs

Ag-In-Ga-S quantum dots

CIGSSe QDs

CuIn0.1Ga0.9(S0.9Se0.1)2 quantum dots

Yb-CsPbI3 QDs

Yb atoms doped in CsPbI3 quantum dots

μGR-CsPbI3 QDs

µ-graphene crosslink CsPbI3 quantum dots

[Pmim]I-CsPbI3 QDs

1-propyl-3-methylimidazolium iodide ionic liquid into CsPbI3 quantum dots

CsPbCl3:Mn QDs

Mn doped in CsPbCl3 quantum dots

Ce3+-CsPbI3 QDs

Ce3+ doped in CsPbI3 quantum dots

Quantum dots are made of toxic metals. This point is not evoked and could constitute a limitation for future developments. Please add a paragraph concerning this point.

Author reply: Thank you very much! According to the valuable suggestion, the research works about the toxicity problem of QDs was consulted and a paragraph concerning this point was added according to this information. (see below)

3.2.3 Other quantum dots as additives in perovskite films

Lead-based halide PSCs are the frontrunner of photovoltaic studies, but the toxic of Pb is a critical bottleneck for the practical use of PSCs. The potential for Pb exposure during the extensive production and operation of PSC technology requires extreme caution due to concerns to the environment and humans. So, it is significant to eliminate or replace Pb with other nontoxic or less toxic elements which can overcome environmental issues.  Among lots of elements, Sn-based PSCs are non-toxic and environment-friendly, but they are criticized for their susceptibility to Sn2+/Sn4+ oxidation, making PSCs less stable and low PCE. There are many ways to inhibit the oxidation of Sn2+/Sn4+ ions, but none of them contain no additives, which can introduce impurities, directly reducing the performance of PSCs. Thus in 2021, Mahmoudi et al. [71] made the first attempt to prepare Sn-based mesoporous PSCs from a mixture of reduced graphene oxide sheets anchored with Sn quantum dots (rGO-Sn QDs) and halogenated tin perovskite in the photoactive layer. This approach effectively inhibited the oxidation of Sn2+/Sn4+ ions, and no impurities were introduced (Figure 6d). The grain size of the rGO-Sn QDs doped film was larger and more compact than the pure film, which can reduce the possibility of water vapor and oxygen entering the device. The final PCE was 7.7%. This work highlighted a new approach to overcome the limitation of environment-friendly alternatives of Pb-based PSCs, which showed a potential for the development of Pb-free PSCs.

  • What about recycling of these solar cells. Please comments.

Author reply: Thank you very much! The cycling stability of specific solar cells has been summarized when discussing specific changes in the photoelectric performance of the cell in the original manuscript. In addition, according to the valuable suggestion, the recycling of these solar cells was also summarized in the revised manuscript. (see below)

  1. Summary and outlook

Fifthly, since lots of PSCs contain toxic Pb, a sustainable procedure for recycling the cells after their operational lifetime is required to prevent exposure of the Pb to environment. In 2016, Binek et al. [162] demonstrated an environmentally responsible and cost-efficient recycling process for MAPbI3-based PSCs. The toxic PbI2 can be recycled, and after recrystallization, can be employed to prepare device, which PCE can reach 13.5%. In addition, they were able to recycle the expensive FTO/glass substrates several times. In 2017, Xu et al. [163] developed in-situ recycling PbI2 from thermal decomposition CH3NH3PbI3 perovskite films for efficient PSCs. This approach can release the risk of Pb outflow. In 2022, Hong et al. [164] investigated novel synthesis of whitlockite (WH,Ca18Mg2(HPO4)2(PO4)12), which can as a high performance Pb2+ absorbent. Their method provided an efficient way to remove Pb2+ ions from water and recycling of PSCs. We should explore more novel approaches to recycle the Pb in PSCs, these will bring one step closer to commercialization.

All of these approaches can be used to optimize the performance of QDs and strengthen the performance of the devices. We are confident that QDs will make a big difference in the field of PSCs.

Reviewer 2 Report

The authors performed a very extensive review of the literature in quantum-sized materials for perovskite solar cells. The language of the manuscript is very appropriate and minor grammatical corrections were highlighted in the attached pdf file.
The application of quantum dots in the absorber layer, additive of or replacing the ETL and HTL was reviewed. Additional recent literature was mentioned in this rapidly evolving field of research. In addition, numerous reports of quantum dots applied as solar cell interlayers, as well as electrodes were reported to improve solar cell performance. A comprehensive review should be inclusive of all reports on the relevant building blocks of solar cells. Detailed comments and literature can be found in the attached pdf file.

Reviewer 3 Report

Good overall work, but with extended problems in English usage, mainly syntactical. Manuscript has been corrected up to page 10. You are advised to incorporate the corrections, and to use them as examples for the rest of the text. If you have access to a professional editor, use his/her services.

Minor corrections in all figures (since most are schematics, you do not need to repeat the term "schematic diagram" etc. in the captions).

You are advised to move all Tables right after the point where they are mentioned in the text. If they come after 5 pages, the reader loses continuity, focus, and interest.

Avoid using abbreviations in the titles of sections. The reader who is skimming the article to find whatever interests him/her, is not necessarily familiar with the abbreviated form. Also, it is better to explain the abbreviation in the first sentences of the subsection.

Round 2

Reviewer 2 Report

The authors applied several revisions and additions to the literature review. As a result, the manuscript is better presented and reflects the current literature more accurately. In the last introductory paragraph, indication of each solar cell layer to each sub-section in the manuscript can greatly help the readership.
There is also suggested literature regarding interlayers and electrodes of solar cells for a comprehensive review of all building blocks of perovskite solar cells, as commented in section 8 of the version 1 of the manuscript.
